# Preference-Based Gradient Estimation for ML-Guided Approximate Combinatorial Optimization

**Arman Mielke**                                                   *arman.mielke@etas.com*
*ETAS Research*
*Computer Science Department, University of Stuttgart*
*Max Planck Research School for Intelligent Systems (IMPRS-IS)*

**Uwe Bauknecht**                                                  *uwe.bauknecht@etas.com*
*ETAS Research*

**Thilo Strauss**                                                  *thilo.strauss@xjtlu.edu.cn*
*School of AI and Advanced Computing, Xi'an Jiaotong-Liverpool University*

**Mathias Niepert**                                          *mathias.niepert@ki.uni-stuttgart.de*
*Computer Science Department, University of Stuttgart*
*NEC Laboratories Europe*
*Max Planck Research School for Intelligent Systems (IMPRS-IS)*

**Reviewed on OpenReview:** *https://openreview.net/forum?id=2S224XC378*

## Abstract

Combinatorial optimization (CO) problems arise across a broad spectrum of domains, including medicine, logistics, and manufacturing. While exact solutions are often computationally infeasible, many practical applications require high-quality solutions within a given time budget. To address this, we propose a learning-based approach that enhances existing non-learned heuristics for CO. Specifically, we parameterize these heuristics and train graph neural networks (GNNs) to predict parameter values that yield near-optimal solutions. Our method is trained end-to-end in a self-supervised fashion, using a novel gradient estimation scheme that treats the heuristic as a black box. This approach combines the strengths of learning and traditional algorithms: the GNN learns from data to guide the algorithm toward better solutions, while the heuristic ensures feasibility. We validate our method on two well-known CO problems: the travelling salesman problem (TSP) and the minimum $k$-cut problem. Our results demonstrate that the proposed approach is competitive with state-of-the-art learned CO solvers.

## 1 Introduction

The design and analysis of heuristics for combinatorial optimization (CO) often focuses on improving worst-case performance. However, such worst-case scenarios may rarely occur in real-world settings. Learning-based approaches offer the advantage of adapting to the distribution of problem instances encountered in practice. Graph neural networks (GNNs) are often the method of choice since most CO problems are either defined on graphs or admit graph-based formulations. Since neural networks cannot directly predict solutions guaranteed to be feasible, generic algorithms such as Monte Carlo tree search and beam search are often used to decode their outputs. However, these methods are impractical to use during training because of their prohibitively long runtime. On the other hand, omitting them during training introduces a discrepancy between training and inference, which can severely impact the optimality gap and generalization.

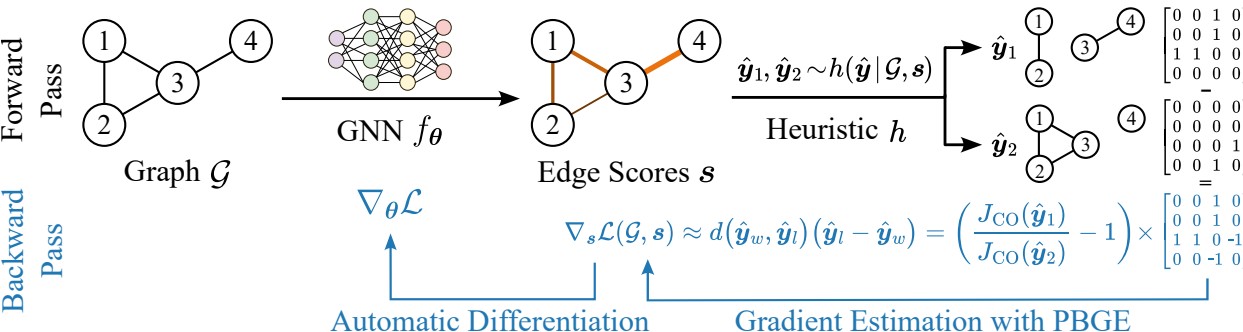

Figure 1: Overview of the proposed framework for ML-guided approximate CO. In the forward pass, the GNN predicts edge scores that are used as input parameters for a CO heuristic. The heuristic operates on these parameters by using them to scale the graph's edge weights and running an off-the-shelf heuristic on the modified graph to obtain a solution $\hat{y}$. Since the CO heuristic is not differentiable in general, a gradient estimation scheme based on preference between a pair of solutions $\hat{y}_1, \hat{y}_2$ is used in the backward pass.

To overcome these limitations, we propose to use GNNs to augment highly efficient heuristics that are commonly available for many CO problems. The GNNs are trained to predict parameters for the heuristics that influence their behavior. Specifically, we focus on edge-selection problems, i.e. problems where the solution is a subset of the edges in the problem graph. The GNN predicts edge weights for the input graph, then we apply the heuristic to this modified graph. By choosing fast heuristics, we can incorporate them into the training loop and use the exact same pipeline during training and inference. Our approach combines the strengths of data-driven methods and established heuristic algorithms: the GNN learns to perform well on frequently encountered instances, and the heuristic, in turn, ensures the feasibility of the resulting solutions.

Once trained, the GNN-augmented heuristic serves as a drop-in replacement for the original heuristic, preserving efficiency and usability while improving solution quality. Indeed, we provide a theoretical guarantee that, for the heuristics that we consider in this work, a suitably trained upstream GNN can learn parameters that render the heuristic optimal.

Since the heuristics return discrete solutions, the gradients of their outputs with respect to their inputs are zero almost everywhere. We therefore need to apply gradient estimation techniques such as the score function and straight-through estimators (Williams, 1992, Bengio et al., 2013) to backpropagate through the algorithms. Since existing gradient estimators cannot be used in a self-supervised setting or have high variance, we propose preference-based gradient estimation (PBGE), which estimates gradients through a comparison of solutions sampled from the given heuristic. This approach enables fully self-supervised training, eliminating the need to pre-compute exact solutions as ground-truth labels, an often costly or even infeasible requirement for many CO problems. Figure 1 illustrates the proposed PBGE framework.

We evaluate our approach on two well-known CO problems, the travelling salesman problem (TSP) and the minimum $k$-cut problem. Our approach improves the performance of a commonly used heuristic by an order of magnitude in the case of minimum $k$-cut while only minimally increasing the runtime. On TSP, our approach is competitive with existing learned approaches and achieves Pareto-optimality.

In summary, our contributions include:

1. A novel gradient estimation scheme, PBGE, for backpropagating through CO heuristics, and

2. An extensive experimental evaluation thereof on common CO problems.

The source code is available at `https://github.com/etas/preference-based-gradient-estimation`.

## 2 Related work

The easiest way of training a model for CO is to assume the existence of ground truth solutions to the CO problems and train in a supervised fashion. Nowak et al. (2017), Joshi et al. (2019) use a GNN to predict an approximate solution as a heatmap, which is then decoded into a feasible solution to the CO problem using beam search. Vinyals et al. (2015) introduce pointer networks, which leverage the fact that many CO problems ask to identify a subset or a permutation of the input. Georgiev et al. (2023) follow a neural algorithmic reasoning approach to learn to imitate CO solvers. Gasse et al. (2019), Kahng et al. (2024) replace components of an existing algorithm that would otherwise be expensive to compute with neural networks. Antoniadis et al. (2025) augment existing algorithms using discrete predictions. Finally, Sun & Yang (2023), Yu et al. (2024) use graph-based denoising diffusion to generate high-quality solutions. However, these supervised approaches aren't applicable to such cases where calculating exact solutions for the training problems is not feasible.

Several approaches have used reinforcement learning (RL) to remove this dependence on a labeled dataset (Kwon et al., 2021, Qiu et al., 2022, Kim et al., 2024). A common approach is to formulate the CO problem as a Markov decision process (Bello et al., 2017, Deudon et al., 2018, Xu et al., 2020, Wu et al., 2022). Khalil et al. (2017), Kool et al. (2019) use the GNN autoregressively to predict which node should be added next to the solution set and repeat that process until a valid solution is reached. Cheng et al. (2023), Pan et al. (2023), McCarty et al. (2021) tackle large-scale instances by locally optimizing sub-parts of the instance individually. However, RL is often sample-inefficient and difficult to train due to the high variance of the gradient estimations.

A common self-supervised approach that does not rely on RL is to formulate a surrogate loss consisting of a term that encourages high quality solutions and one that softly enforces the constraints (Karalias & Loukas, 2020, Sun et al., 2022, Min et al., 2022; 2023, Bu et al., 2024, Wenkel et al., 2024). Duan et al. (2022) train self-supervised using a contrastive loss instead. Schuetz et al. (2022), Sanokowski et al. (2024) focus on quadratic and polynomial unconstrained binary optimization, allowing them to formulate self-supervised loss functions for these problem families. Tönshoff et al. (2021) use an LSTM-based architecture to solve binary maximum constraint satisfaction problems, which many CO problems can be formulated as. Corsini et al. (2024) train a pointer network-based model in a self-supervised fashion. Zhang et al. (2023) tackle CO using GFlowNets (Bengio et al., 2021). Yau et al. (2024) propose a GNN architecture that can capture message passing algorithms with provable approximation guarantees for a large class of CO problems. Pan et al. (2025), Liao et al. (2025) also apply preference learning to CO; Pan et al. (2025) decode solutions by modelling the CO problem as a Markov decision process and constructing the solution autoregressively, while Liao et al. (2025) use a sampling decoder and thin out the solution pool while retaining a range of solutions of different quality.

Joshi et al. (2021) compare some of the paradigms introduced in other papers in structured experiments. Xia et al. (2024) argue that the common pattern of training the GNN using a surrogate loss, but testing it using a decoder, means that the GNN is applied inconsistently, leading to uncertain performance during testing. Our approach addresses this problem by incorporating the decoder during training. There have been two lines of work on backpropagating through CO problems. Firstly, if we have a set of optimal solutions given as training data, we can use supervised learning to train the GNN to output adjacency matrices as close as possible to the optimal solutions (Elmachtoub & Grigas, 2022). This is often called "predict, then optimize". Secondly, there are several methods to backpropagate through a non-differentiable CO algorithm, such as Niepert et al. (2021), Minervini et al. (2023), Vlastelica et al. (2020). Related to our work is decision-focused learning, which has developed several methods to backpropagate through CO solvers (Mandi et al., 2024). Our approach follows this paradigm. The implicit maximum likelihood estimator (I-MLE) (Niepert et al., 2021) is another such method; we compare against it in our experiments and show that our approach improves over I-MLE. Preference learning has previously been used outside CO for fine-tuning large language models (LLMs) (Badrinath et al., 2024, Rafailov et al., 2023, Meng et al., 2024). For instance, Badrinath et al. (2024) propose a hybrid approach between preference learning and reinforcement learning from human feedback.

## 3 Background

**CO problems.** A combinatorial optimization (CO) problem asks us, given a discrete set $M$ and an objective function $J_{\mathrm{CO}} : M \to \mathbb{R}$, to find the minimum

$$\min_{x \in M} J_{\mathrm{CO}}(x).$$

Maximization problems can be turned into minimization problems by inverting the sign of the objective function. Since finding the exact global optimum is often not required in practice, this paper focuses on efficiently finding approximate solutions.

To illustrate our approach, we will refer to specific CO problems as follows. The minimum $k$-cut problem asks, given a weighted, undirected graph and a desired number $k$ of connected components, to find a set of edges with minimum total weight whose removal leaves the graph with exactly $k$ connected components. A commonly used heuristic for this problem is the Karger–Stein algorithm (Karger & Stein, 1993). In the travelling salesman problem (TSP), we are given a weighted, undirected graph, and are asked to find a minimum-weight Hamiltonian cycle, i.e. a cycle that visits every node exactly once and where the sum of the weights of edges that are traversed in the cycle is minimal. A well-known probabilistic heuristic for solving the TSP is the random insertion algorithm (Karg & Thompson, 1964). The variant of the TSP most commonly experimented on in related literature (Kool et al., 2019, Joshi et al., 2019; 2021) is the Euclidean TSP, where the graph is fully connected, the nodes represent points in the unit square and the edge weights are the Euclidean distances between these points. We give formal definitions for both problems as well as descriptions of the algorithms mentioned in Appendix A.

**Graph neural networks (GNNs).** Let $\mathcal{G} = (V, E)$ be a graph with $n = |V|$ the number of nodes. Let $\mathbf{X} \in \mathbb{R}^{n \times d}$ be the feature matrix that associates each node of the graph with a $d$-dimensional feature vector and let $\mathbf{A} \in \mathbb{R}^{n \times n}$ be the adjacency matrix. GNNs based on the message passing paradigm have three basic operations, which are defined as

$$\mathbf{h}_i^{\ell} = \gamma \left( \mathbf{h}_i^{\ell-1}, \varphi_{j \in \mathcal{N}(v_i)} \phi \left( \mathbf{h}_i^{\ell-1}, \mathbf{h}_j^{\ell-1}, r_{ij} \right) \right),$$

where $\gamma$, $\varphi$, and $\phi$ represent update, aggregation and message function respectively.

**Propagation step.** The message-passing network computes a message $m_{ij}^{\ell} = \phi(\mathbf{h}_i^{\ell-1}, \mathbf{h}_j^{\ell-1}, r_{ij})$ between every pair of neighboring nodes $(v_i, v_j)$. The function takes in input $v_i$'s and $v_j$'s representations $\mathbf{h}_i^{\ell-1}$ and $\mathbf{h}_j^{\ell-1}$ at the previous layer $\ell - 1$, and the relation $r_{ij}$ between the two nodes.

**Aggregation step.** For each node in the graph, the network performs an aggregation computation over the messages from $v_i$'s neighborhood $\mathcal{N}(v_i)$ to calculate an aggregated message $M_i^{\ell} = \varphi(\{m_{ij}^{\ell} \mid v_j \in \mathcal{N}(v_i)\})$. The definition of the aggregation function differs between methods.

**Update step.** Finally, the model non-linearly transforms the aggregated message $M_i^{\ell}$ and $v_i$'s representation from previous layer $\mathbf{h}_i^{\ell-1}$ to obtain $v_i$'s representation at layer $\ell$ as $\mathbf{h}_i^{\ell} = \gamma(\mathbf{h}_i^{\ell-1}, M_i^{\ell})$.

**Gradient Estimators for Discrete Distributions.** Learning systems that backpropagate through discrete distributions and algorithms require gradient estimation approaches. This is because gradients cannot be directly computed or are zero almost everywhere for operations such as sampling from a categorical distribution or running a solver for CO problems. The standard estimator is the score function estimator, also known as the REINFORCE estimator (Williams, 1992). Given a function $J$ and a parameterized probability distribution $p_\theta(x)$ it estimates the true gradient as follows:

$$\nabla_\theta \mathbb{E}_{y \sim p_\theta(x)} \big[ J(y) \big] = \mathbb{E}_{y \sim p_\theta(x)} \big[ J(y) \nabla_\theta \log p_\theta(x) \big] \approx \frac{1}{S} \sum_{i=1}^{S} J(y_i) \nabla_\theta \log p_\theta(x_i), \quad y_i \sim p_\theta(x_i),$$

where $S$ is the number of samples used to estimate the expectation. While this estimator is unbiased, it suffers from large variance. There are other estimators, such as the straight-through estimator (Bengio et al., 2013) and I-MLE (Niepert et al., 2021) with smaller variance but biased gradient estimates.

## 4 Problem statement

We consider CO problems on graphs with a linear objective function $J_{\text{CO}}(\hat{\boldsymbol{y}}) = \langle \hat{\boldsymbol{y}}, \boldsymbol{c} \rangle$ for some constant $\boldsymbol{c} \in \mathbb{R}^{|E|}$, and a probabilistic heuristic. The heuristic takes as input a graph $\mathcal{G} = (V, E, w)$ with nodes $V$, edges $E$ and edge weights $w : E \to \mathbb{R}_{>0}$, and returns (samples) a potentially suboptimal solution $\hat{\boldsymbol{y}}$. The heuristic, therefore, defines a probability distribution $h(\hat{\boldsymbol{y}} \mid \mathcal{G})$ over the solutions it outputs for a given graph $\mathcal{G}$. For instance, we might have the minimum $k$-cut problem using the Karger–Stein algorithm as the heuristic.

We now want to use a GNN $f_{\boldsymbol{\theta}}$ parameterized by $\boldsymbol{\theta}$ applied to the input graphs $\mathcal{G}$ to compute an updated graph $\mathcal{G}' = f_{\boldsymbol{\theta}}(\mathcal{G})$ such that the probabilistic heuristic when applied to this new graph is improved in expectation. Hence, we want to solve the following optimization problem:

$$\min_{\boldsymbol{\theta}} \mathbb{E}_{\hat{\boldsymbol{y}} \sim h(\hat{\boldsymbol{y}}|f_{\boldsymbol{\theta}}(\mathcal{G}))} \big[ J_{\text{CO}}(\hat{\boldsymbol{y}}) \big].$$

For each input graph $\mathcal{G}$, $h(\hat{\boldsymbol{y}} \mid f_{\boldsymbol{\theta}}(\mathcal{G}))$ is a discrete probability distribution (due to the assumption that $h$ is probabilistic) parameterized by $\boldsymbol{\theta}$. The main challenge in solving this optimization problem is that (discrete) heuristics are typically not differentiable functions and that optimal solutions are prohibitively expensive to obtain as training data, making supervised training infeasible. Moreover, we assume that the heuristic is a black box—while we can sample from the probability distribution defined by it, we cannot compute a probability mass for a given sample.

## 5 Method

We introduce a novel approach to CO by deriving a new gradient estimator that enables backpropagation through probabilistic heuristics. This allows us, for the first time, to directly leverage solution quality rankings, which are automatically generated via objective function-based comparisons, to learn to improve CO heuristics. We use GNNs to guide existing probabilistic heuristics for a given CO problem. The GNN receives the problem graph as input and produces a prior score for each edge. These scores are used as additional input alongside the graph for a parameterized version of an off-the-shelf CO heuristic, which then produces a solution to the CO problem. Figure 1 shows an overview of our approach.

Since the heuristic is not differentiable in general, we use gradient estimation to obtain the gradients with respect to the GNN's output. Existing gradient estimation schemes, such as REINFORCE, the straight-through estimator, Gumbel softmax, or I-MLE, either exhibit high bias or variance or require a differentiable loss function. We instead propose a new gradient estimation scheme based on preference-based optimization, which we term preference-based gradient estimation (PBGE).

### 5.1 Parameterizing heuristics

A heuristic that takes a problem graph as input can be indirectly parameterized by modifying the graph before executing the algorithm; the modification applied to the graph influences the behavior and output of the heuristic. We modify an input graph $\mathcal{G}$ by using the GNN's output to change its edge weights. Assume there is an arbitrary but fixed ordering of edges. The model outputs prior scores $\boldsymbol{s} = f_{\boldsymbol{\theta}}(\mathcal{G}) \in \mathbb{R}^{|E|}$, one score per edge. A high score for a given edge is interpreted to mean that the respective edge should belong to the solution set with a higher probability mass. The heuristics we use prefer including edges of low weight in the solution set, therefore we scale down the weights of edges that received high scores. Specifically, the edge weights are multiplied with $1 - \sigma(\boldsymbol{s})$, where $\sigma$ is the element-wise sigmoid function. By running the CO heuristic on this modified graph, we parameterize the heuristic using the GNN's output scores $\boldsymbol{s}$. Importantly, since the heuristic guarantees that its output satisfies the constraints of the CO problem, the same is also true for the parameterized heuristic. In the remainder of this paper, $h(\hat{\boldsymbol{y}} \mid \mathcal{G}, \boldsymbol{s}) = h(\hat{\boldsymbol{y}} \mid \mathcal{G}')$ denotes the probability distribution defined by a probabilistic CO heuristic parameterized in this way. It samples and outputs a vector $\hat{\boldsymbol{y}} \in \{0, 1\}^{|E|}$ that represents a solution to the CO problem, such as a TSP tour or $k$-cut. A value of 1 in $\hat{\boldsymbol{y}}$ means that the corresponding edge is in the solution set.

## 5.2 Preference-based gradient estimation (PBGE)

In preference learning, a training instance consists of an input and a pair of possible outputs. The supervision signal is an annotation indicating that one of the outputs $\boldsymbol{y}_w$ is of higher quality than the other output $\boldsymbol{y}_l$. We can construct a similar setup for CO by leveraging a pre-existing probabilistic CO heuristic $h$. Sampling from the heuristic multiple times likely yields two solutions $\hat{\boldsymbol{y}}_1, \hat{\boldsymbol{y}}_2 \sim h(\hat{\boldsymbol{y}} \mid \mathcal{G}, \boldsymbol{s}')$ of different quality for a given problem instance $\mathcal{G} \sim \mathcal{D}$ from dataset $\mathcal{D}$. Here we use $\boldsymbol{s}' = f_{\boldsymbol{\theta}'}(\mathcal{G})$ obtained from the previous training iteration's model. These solutions can easily be ranked by applying the CO problem's objective function[1] $J_{\mathrm{CO}}$. This means assigning $\hat{\boldsymbol{y}}_w$ (winner) and $\hat{\boldsymbol{y}}_l$ (loser) such that $J_{\mathrm{CO}}(\hat{\boldsymbol{y}}_w) \leq J_{\mathrm{CO}}(\hat{\boldsymbol{y}}_l)$.

We now propose the following preference-based loss function:

$$\mathcal{L}(\mathcal{D}, \boldsymbol{s}) = \mathbb{E}_{\hat{\boldsymbol{y}}_w, \hat{\boldsymbol{y}}_l \sim h(\hat{\boldsymbol{y}}|\mathcal{G}, \boldsymbol{s}'), \, \mathcal{G} \sim \mathcal{D}} \left[ d(\hat{\boldsymbol{y}}_w, \hat{\boldsymbol{y}}_l) \log \left( \frac{h(\hat{\boldsymbol{y}}_l \mid \mathcal{G}, \boldsymbol{s})}{h(\hat{\boldsymbol{y}}_w \mid \mathcal{G}, \boldsymbol{s})} \right) \right]. \tag{1}$$

Intuitively, this rewards assigning a high probability to the better solution $\hat{\boldsymbol{y}}_w$ and a low probability to the worse solution $\hat{\boldsymbol{y}}_l$. $d(\hat{\boldsymbol{y}}_w, \hat{\boldsymbol{y}}_l)$ is a scaling factor;[2] as we will see later, its purpose is to scale the gradients based on the distance between the objective values of $\hat{\boldsymbol{y}}_w$ and $\hat{\boldsymbol{y}}_l$.

Since we treat the CO heuristic as a black box, we cannot calculate the probabilities $h(\hat{\boldsymbol{y}}_w \mid \mathcal{G}, \boldsymbol{s})$ and $h(\hat{\boldsymbol{y}}_l \mid \mathcal{G}, \boldsymbol{s})$ directly. We therefore introduce a proxy distribution $\pi(\hat{\boldsymbol{y}} \mid \mathcal{G}, \boldsymbol{s}) \approx h(\hat{\boldsymbol{y}} \mid \mathcal{G}, \boldsymbol{s})$ for which we can obtain probabilities directly. For all heuristics used in this paper, a high prior score in $\boldsymbol{s}$ for a certain edge increases the probability of this edge being included in the output $\hat{\boldsymbol{y}}$. This motivates the use of an exponential family distribution to model the proxy distribution $\pi$ for $h$:

$$\pi(\hat{\boldsymbol{y}} \mid \mathcal{G}, \boldsymbol{s}) = \frac{\exp(\langle \hat{\boldsymbol{y}}, \boldsymbol{s} \rangle)}{\sum_{\boldsymbol{y}' \in \mathcal{C}} \exp(\langle \boldsymbol{y}', \boldsymbol{s} \rangle)}, \tag{2}$$

where $\langle \cdot, \cdot \rangle$ is the inner product and $\mathcal{C}$ is the set of all solutions to the CO problem. Since the CO problem's objective function $J_{\mathrm{CO}}$ is linear, $\pi$ models a distribution proportional to $J_{\mathrm{CO}}$. On top of this, the numerator of $\pi$ is easily differentiable, which we can use to estimate the gradient of the loss.

Replacing $h$ with $\pi$ in Equation 1 and inserting Equation 2 simplifies the loss function to

$$\mathcal{L}(\mathcal{D}, \boldsymbol{s}) = \mathbb{E}_{\hat{\boldsymbol{y}}_w, \hat{\boldsymbol{y}}_l \sim h(\hat{\boldsymbol{y}}|\mathcal{G}, \boldsymbol{s}'), \, \mathcal{G} \sim D} \left[ d(\hat{\boldsymbol{y}}_w, \hat{\boldsymbol{y}}_l) \big( \langle \hat{\boldsymbol{y}}_l, \boldsymbol{s} \rangle - \langle \hat{\boldsymbol{y}}_w, \boldsymbol{s} \rangle \big) \right]. \tag{3}$$

Now, the gradient of this expectation with respect to $\boldsymbol{s}$ is

$$\nabla_{\boldsymbol{s}} \mathcal{L}(\mathcal{D}, \boldsymbol{s}) = \mathbb{E}_{\hat{\boldsymbol{y}}_w, \hat{\boldsymbol{y}}_l \sim h(\hat{\boldsymbol{y}}|\mathcal{G}, \boldsymbol{s}'), \, \mathcal{G} \sim D} \left[ d(\hat{\boldsymbol{y}}_w, \hat{\boldsymbol{y}}_l)(\hat{\boldsymbol{y}}_l - \hat{\boldsymbol{y}}_w) \right], \tag{4}$$

whose single-sample Monte Carlo estimate can be written as

$$\nabla_{\boldsymbol{s}} \mathcal{L}(\mathcal{G}, \boldsymbol{s}) \approx d(\hat{\boldsymbol{y}}_w, \hat{\boldsymbol{y}}_l)(\hat{\boldsymbol{y}}_l - \hat{\boldsymbol{y}}_w), \quad \text{where } \hat{\boldsymbol{y}}_w, \hat{\boldsymbol{y}}_l \sim h(\hat{\boldsymbol{y}} \mid \mathcal{G}, \boldsymbol{s}'). \tag{5}$$

Intuitively, the gradient is negative at a certain edge if that edge is in the better solution, but not in the worse solution. A negative gradient raises the GNN's output score, meaning that the GNN will be nudged towards including this edge in its solution. Similarly, a positive gradient means that the corresponding edge was in the worse solution, but not in the better solution. This nudges the GNN's output down, so it pushes the GNN towards not including this edge.

The form of the gradient is reminiscent of those used for preference learning with LLMs (Rafailov et al., 2023, Meng et al., 2024). There, the gradient makes the model increase the likelihood of the better solution and decrease the likelihood of the worse solution, and a scaling factor is used to weigh important gradients more highly. Unlike the preference learning setting used with LLMs, we not only know which solution in a pair is better, but we can measure the quality of each solution exactly using the objective function. This eliminates the need for human annotators to rank pairs of examples. Moreover, we can leverage the objective function to more easily compute a suitable scaling factor.

---

[1]While $J_{\mathrm{CO}}$ depends on graph $\mathcal{G}$ and predicted solution $\hat{\boldsymbol{y}}$, we omit the graph parameter for readability.

[2]The input graph $\mathcal{G}$ is also an argument of $d$, but we omit it for ease of readability.

**Gradient scaling.** If the solutions $\hat{\boldsymbol{y}}_w$ and $\hat{\boldsymbol{y}}_l$ are of similar quality, we do not want to strongly move the GNN towards either solution. We therefore scale the gradient with the relative optimality gap between the two solutions,

$$d(\hat{\boldsymbol{y}}_w, \hat{\boldsymbol{y}}_l) = \frac{J_{\text{CO}}(\hat{\boldsymbol{y}}_l)}{J_{\text{CO}}(\hat{\boldsymbol{y}}_w)} - 1.$$

This is always non-negative, since, by definition, $J_{\text{CO}}(\hat{\boldsymbol{y}}_l) \geq J_{\text{CO}}(\hat{\boldsymbol{y}}_w)$. Using this scaling factor places more weight on pairs of solutions where the difference in their respective optimality gaps is large. In particular, due to the $-1$ term, if the two solutions are of the same quality, the gradient is set to zero. This means we do not move the GNN towards either solution. The scaled gradient is

$$\nabla_{\boldsymbol{s}}\mathcal{L}(\mathcal{G}, \boldsymbol{s}) \approx \left( \frac{J_{\text{CO}}(\hat{\boldsymbol{y}}_l)}{J_{\text{CO}}(\hat{\boldsymbol{y}}_w)} - 1 \right) (\hat{\boldsymbol{y}}_l - \hat{\boldsymbol{y}}_w).$$

**Solution pool.** The variance of the gradient can be reduced by estimating the expectation in Equation 4 by creating a pool of solutions with the heuristic and by constructing pairs from this pool. We form the pairs by combining the best solution from the pool with each of the weaker solutions. In practice, the accuracy of the gradients depends heavily on the quality of the best found solution $\hat{\boldsymbol{y}}_w$. At the beginning of training, the GNN cannot yet output good enough scores to consistently find reasonable $\hat{\boldsymbol{y}}_w$. To remedy this, we also run the heuristic on the unmodified graph and add the resulting solutions to the pool from which the pairs are generated. This has the additional benefit of ensuring that multiple distinct solutions can always occur in the solution pool, even if the GNN outputs scores that strongly favor one particular solution, thus preventing training from stalling. The complete training procedure is described in Algorithm 1.

---

**Algorithm 1** One training iteration with PBGE

$\boldsymbol{s'} \leftarrow f_{\boldsymbol{\theta'}}(\mathcal{G})$

▷ Sample $n$ solutions from $h$ guided by the current $\boldsymbol{s'}$, and $m$ unguided solutions

$\hat{\boldsymbol{y}}_1, \ldots, \hat{\boldsymbol{y}}_n \sim h(\hat{\boldsymbol{y}} \mid \mathcal{G}, \boldsymbol{s'})$

$\hat{\boldsymbol{y}}_{n+1}, \ldots, \hat{\boldsymbol{y}}_{n+m} \sim h(\hat{\boldsymbol{y}} \mid \mathcal{G})$

$\hat{\mathcal{Y}} \leftarrow \{\hat{\boldsymbol{y}}_1, \ldots, \hat{\boldsymbol{y}}_{n+m}\}$

$\hat{\boldsymbol{y}}_w \leftarrow \arg\min_{\hat{\boldsymbol{y}} \in \hat{\mathcal{Y}}} J_{\text{CO}}(\hat{\boldsymbol{y}})$

▷ Estimate gradient using each pair involving $\hat{\boldsymbol{y}}_w$

$$\nabla_{\boldsymbol{s}}\mathcal{L}(\mathcal{G}, \boldsymbol{s}) \approx \sum_{\hat{\boldsymbol{y}}_l \in \hat{\mathcal{Y}}} \left( \frac{J_{\text{CO}}(\hat{\boldsymbol{y}}_l)}{J_{\text{CO}}(\hat{\boldsymbol{y}}_w)} - 1 \right) (\hat{\boldsymbol{y}}_l - \hat{\boldsymbol{y}}_w)$$

Backpropagate gradient $\nabla_{\boldsymbol{s}}\mathcal{L}(\mathcal{G}, \boldsymbol{s})$

---

**Decoding at test time.** At test time, the model's output needs to be converted (decoded) to a solution to the CO problem. This can simply be done by running the CO heuristic with the model's output as input, as described in Section 4. The solution can be improved by running a probabilistic CO heuristic repeatedly and using the best solution found as final output.

### 5.3 Theoretical analysis

We provide the following theoretical result, which illustrates that we can turn a probabilistic CO heuristic into an exact algorithm if we find an optimal modified input graph.

> **Theorem 1.** *Let $\mathcal{G} = (V, E, w)$ be an undirected, weighted graph. Moreover, let $h(\hat{\boldsymbol{y}} \mid \mathcal{G}, \boldsymbol{s})$ be the probability distribution defined by one of the following heuristics, guided by parameters $\boldsymbol{s} \in \mathbb{R}^{|E|}$:*
>
> - *the Karger–Stein algorithm for the minimum $k$-cut problem,*
>
> - *an insertion algorithm for the TSP, e.g. random insertion, or*
>
> - *the (deterministic) Christofides algorithm for the TSP.*
>
> *Finally, let $\boldsymbol{y}$ be an optimal solution for the respective problem on $\mathcal{G}$. There exists a set of parameters $\boldsymbol{s} \in \mathbb{R}^{|E|}$ such that*
> $$h(\boldsymbol{y} \mid \mathcal{G}, \boldsymbol{s}) = 1.$$

In other words, there are GNN output parameters that *guarantee* that an optimal solution is found, even if a randomized heuristic such as Karger–Stein or random insertion is used. In Appendix B we break this theorem down for each individual algorithm and provide more rigorous formulations as well as proofs.

## 6 Experiments

We validate our approach on two well-known CO problems: the TSP and the minimum $k$-cut problem. For both problems, we synthetically generate problem instances and establish baselines as reference. We use residual gated graph convnets (Bresson & Laurent, 2017), but adapt them to include edge features $\boldsymbol{e}_{ij}^l$ and a dense attention map $\boldsymbol{\eta}_{ij}^l$ following Joshi et al. (2019). Please see Appendix C for details.

The selection of the probabilistic heuristic for PBGE should be made with the trade-off between solution quality and runtime in mind. The algorithm needs to be fast enough to be run during training, while simultaneously delivering good enough solutions for PBGE to construct useful gradients. For the minimum $k$-cut problem, we use the Karger–Stein algorithm as our heuristic, as it is considered optimal in terms of the probability of finding the minimum $k$-cut (Gupta et al., 2020), and runs quickly enough to be used during training. For the TSP, we choose the random insertion heuristic; while higher-quality heuristics exist, their longer runtime prevents them from being useful during training.

Table 1: Minimum $k$-cut optimality gaps on graphs with 100 nodes and $k = 2$, using Karger–Stein as decoder. Mean $\pm$ standard deviation were calculated over ten evaluation runs on the same model parameters. In the supervised and self-supervised rows, a GNN trained with the indicated method is used to guide the Karger–Stein algorithm. In the columns labeled "Best out of 3 runs", the Karger–Stein algorithm is run three times on the same GNN outputs, and the best result is used.

| Method | Unweighted graphs | | NOIgen+ | |
|---|---|---|---|---|
| | Single run | Best out of 3 runs | Single run | Best out of 3 runs |
| **Non-learned** | | | | |
| Karger–Stein | 3.61%±0.21 (330ms) | 0.62%±0.07 (971ms) | 11.29%±0.71 (352ms) | 0.43%±0.11 (1.00s) |
| **Supervised** | | | | |
| BCE loss | 0.27%±0.06 (402ms) | 0.03%±0.02 (1.05s) | 0.41%±0.07 (415ms) | 0.06%±0.04 (1.04s) |
| I-MLE | 1.67%±0.10 (415ms) | 0.14%±0.06 (1.03s) | 2.53%±0.12 (435ms) | 0.28%±0.06 (1.08s) |
| **Self-supervised** | | | | |
| REINFORCE | 2.18%±0.15 (408ms) | 0.41%±0.11 (1.03s) | 3.15%±0.36 (440ms) | 0.18%±0.03 (1.09s) |
| I-MLE | 3.39%±0.16 (402ms) | 0.52%±0.05 (1.01s) | 7.63%±0.41 (444ms) | 0.41%±0.07 (1.10s) |
| Id. | 3.62%±0.15 (407ms) | 0.61%±0.05 (1.02s) | 7.46%±0.40 (428ms) | 0.42%±0.07 (1.06s) |
| PBGE (**ours**) | 0.38%±0.05 (398ms) | 0.06%±0.05 (1.01s) | 0.58%±0.06 (439ms) | 0.09%±0.05 (1.08s) |

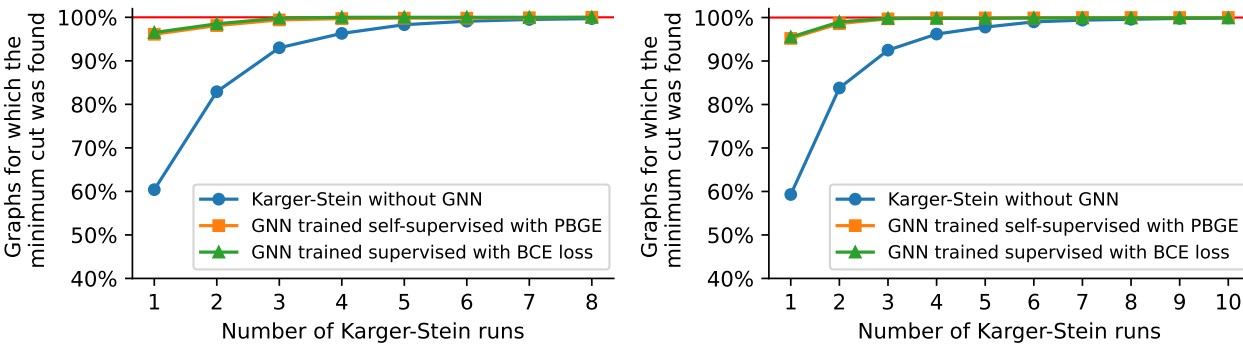

(a) On graphs without edge weights and 100 nodes.   (b) On graphs without edge weights and 200 nodes.

Figure 2: The number of graphs for which the minimum $k$-cut was found after a given number of Karger–Stein runs, on $k = 2$. For example, for Karger–Stein without a GNN on graphs with 100 nodes (Figure 2a), two Karger–Stein runs suffice to find the minimum $k$-cut for 83% of graphs in the validation set.

Table 2: Minimum $k$-cut optimality gaps on unweighted graphs with 100 nodes, for $k = 3$ and $k = 4$. The row labeled "PBGE" refers to a GNN trained self-supervised with PBGE, using the Karger–Stein algorithm as decoder. In the columns labeled "Best out of 3 runs", the Karger–Stein algorithm is run three times on the same GNN outputs, and the best result is used.

| **Method** | $k = 3$ | | $k = 4$ | |
| | Single run | Best out of 3 runs | Single run | Best out of 3 runs |
| --- | --- | --- | --- | --- |
| Karger–Stein | 32.95% | 4.93% | 45.92% | 9.16% |
| PBGE (**ours**) | 3.71% | 0.04% | 9.72% | 0.81% |

### 6.1 Problem instance generation and baselines

For minimum $k$-cut, we use the established graph generator *NOIgen* (Nagamochi et al., 1994). Since it relies on dramatically scaling down the weights of edges that are in the minimum $k$-cut in order to avoid trivial solutions, it makes it easy for a GNN to identify the correct edges. To make the graphs more challenging, we extend NOIgen to also use graph structure to avoid trivial solutions, which allows us to scale down edge weights less dramatically. We call this improved graph generator *NOIgen+*. We also generate unweighted graphs that only rely on graph structure to prevent trivial solutions. Graphs for the TSP are generated according to the established method described in Kool et al. (2019), Joshi et al. (2019; 2021). These datasets are named TSP-20, TSP-50 and TSP-100 according to the number of nodes in each graph. For each dataset, we generate 10,000 training graphs, 1,000 validation graphs and 1,000 test graphs. Please refer to Appendix E for further details.

Since we assume a setting without access to ground truth solutions to the CO problems, our primary baselines are gradient estimation schemes for unsupervised training. We set the loss to

$$\mathcal{L}(\mathcal{D}, \boldsymbol{s}) = \mathbb{E}_{\hat{\boldsymbol{y}} \sim h(\hat{\boldsymbol{y}} | \mathcal{G}, \boldsymbol{s}), \, \mathcal{G} \sim D} \big[ J_{\mathrm{CO}}(\hat{\boldsymbol{y}}) \big]$$

and estimate $\nabla_{\boldsymbol{s}} \mathcal{L}(\mathcal{D}, \boldsymbol{s})$ using I-MLE (Niepert et al., 2021), and REINFORCE (Williams, 1992), as well as identity with projection (Sahoo et al., 2023), which we refer to as "Id." in our tables. Here, I-MLE and identity with projection act as representatives of the decision-focused learning paradigm (Mandi et al., 2024). We note that other approaches from this field, such as Berthet et al. (2020), can also be applied here. We also train supervised baselines using an edge-level binary cross entropy (BCE) loss comparing the GNN's output scores $\boldsymbol{s}$ with a pre-calculated ground truth solution $\boldsymbol{y}$. Additionally, we train models using I-MLE in a supervised fashion, comparing the heuristic's output $\hat{\boldsymbol{y}}$ with $\boldsymbol{y}$ using a Hamming loss. See Appendix D for details on these baselines.

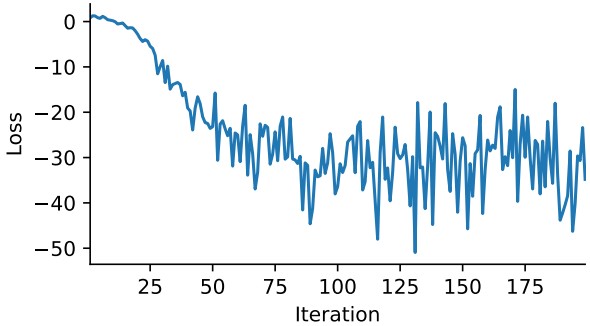

Figure 3: Evolution of the unscaled loss while training on minimum $k$-cut with $k = 2$, using unweighted graphs with 100 nodes.

Table 3: The amount of time required to train a GNN for minimum $k$-cut on unweighted graphs using PBGE, for different graph sizes ($n$) and values of $k$.

|  | $k = 2$ | $k = 3$ | $k = 4$ |
|---|---|---|---|
| $n = 100$ | 4h 8min | 7h 39min | 6h 13min |
| $n = 200$ | 15h 46min | - | - |

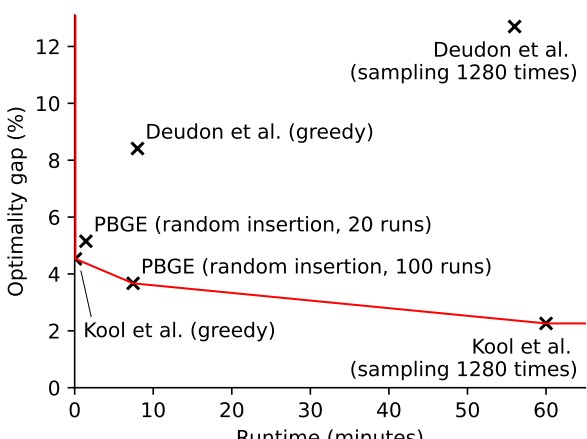

Figure 4: The Pareto frontier of self-supervised methods on TSP-100. Non-Pareto-optimal baselines with very high optimality gaps or very long runtimes are omitted for clarity.

## 6.2 Results

**Minimum $k$-Cut.** We evaluate our method on the minimum $k$-cut problem, using the Karger–Stein algorithm as a base. Table 1 shows optimality gaps of the unmodified Karger–Stein algorithm, as well as several versions of our method. Each version augments the Karger–Stein algorithm with a GNN, and they differ by how the GNN was trained. In practice, it is common to run the Karger–Stein algorithm multiple times, and to use the best found cut as the final result. We therefore report optimality gaps after decoding using a single Karger–Stein run, as well as decoding with 3 Karger–Stein runs on the same GNN outputs. Note that when augmenting the Karger–Stein algorithm with a GNN trained with PBGE, the optimality gap improves by an order of magnitude compared to the unmodified Karger–Stein algorithm. On top of this, the GNN trained with PBGE comes close to matching the GNN trained supervised with a BCE loss. This is despite the fact that the BCE loss assumes access to ground truth solutions for supervised training, which leads to a much easier setting.

In practice, the most important metric is the number of runs for Karger–Stein to find the optimal $k$-cut. If this number is low, we can run Karger–Stein a small number of times and be reasonably certain that the minimum $k$-cut was found. Figure 2 shows for how many graphs the minimum $k$-cut is found in a set number of runs, comparing the unmodified Karger–Stein algorithm with two versions that were augmented using a GNN. On both datasets, the augmented Karger–Stein algorithm needs much fewer runs to find the minimum $k$-cut, almost always finding it on the first attempt. Again, the GNN trained self-supervised with PBGE comes close to matching supervised performance.

The results so far have used $k = 2$, but our approach can be applied to any value of $k$. Table 2 shows optimality gaps of PBGE and Karger–Stein with $k = 3$ and $k = 4$. In both cases, augmenting the Karger–Stein algorithm using a GNN trained self-supervised with PBGE substantially improves the optimality gap. Figure 3 shows how the (unscaled) loss in Equation 3 evolves during training. The figure demonstrates that the loss decreases until training converges, indicating that the PBGE update rule optimizes the proposed objective. Note that we only calculate the loss for logging purposes here; since we train using the estimated gradient, the loss generally does not need to be computed explicitly. We show an overview of the amount of time required to train using PBGE for different values of $n$ and $k$ in Table 3. Additional experiments can be found in Appendix F.1.

Table 4: TSP optimality gaps, with mean ± standard deviation, calculated over ten evaluation runs on the same model parameters. We indicate the decoder used in parentheses. Values with * are obtained from these papers and therefore do not include standard deviations. We provide an extended version of this table with more baselines, including supervised BCE and I-MLE, in Appendix F.2.

| Method (decoder in parentheses) | $n = 20$ | $n = 50$ | $n = 100$ |
|---|---|---|---|
| **Self-supervised** | | | |
| Bello et al. (greedy) | 1.42%* | 4.46%* | 6.90%* |
| Khalil et al. (greedy) | 1.42%* | 5.16%* | 7.03%* |
| Deudon et al. (greedy) | 0.66%* (2m) | 3.98%* (5m) | 8.41%* (8m) |
| Deudon et al. (greedy + 2OPT) | 0.42%* (4m) | 2.77%* (26m) | 5.21%* (3h) |
| Deudon et al. (sampling 1280 times) | 0.11%* (5m) | 1.28%* (17m) | 12.70%* (56m) |
| Deudon et al. (sampling 1280 times + 2OPT) | 0.09%* (6m) | 1.00%* (32m) | 4.64%* (5h) |
| Kool et al. (greedy) | 0.34%* | 1.76%* (2s) | 4.53%* (6s) |
| Kool et al. (sampling 1280 times) | 0.08%* (5m) | 0.52%* (24m) | 2.26%* (1h) |
| REINFORCE (random ins., 20 runs) | 7.98%±0.08 (769ms) | 23.84%±0.10 (11.62s) | 52.83%±0.47 (1.38m) |
| REINFORCE (random ins., 100 runs) | 4.78%±0.03 (3.64s) | 11.24%±0.08 (57.85s) | 31.06%±0.25 (7.53m) |
| I-MLE (random ins., 20 runs) | 10.54%±0.15 (761ms) | 35.87%±0.34 (11.14s) | 61.42%±0.23 (1.37m) |
| I-MLE (random ins., 100 runs) | 6.55%±0.10 (3.68s) | 19.40%±0.24 (56.63s) | 44.42%±0.14 (7.51m) |
| Id. (random ins., 20 runs) | 8.92%±0.13 (762ms) | 31.87%±0.32 (11.10s) | 56.55%±0.33 (1.41m) |
| Id. (random ins., 100 runs) | 5.16%±0.09 (3.66s) | 17.85%±0.09 (55.92s) | 39.69%±0.28 (7.48m) |
| PBGE **(ours)** (random ins., 20 runs) | 0.18%±0.01 (763ms) | 2.37%±0.02 (11.11s) | 5.12%±0.06 (1.43m) |
| PBGE **(ours)** (random ins., 100 runs) | 0.05%±0.01 (3.73s) | 1.13%±0.03 (53.98s) | 3.67%±0.05 (7.44m) |
| **Non-learned heuristics** | | | |
| Christofides | 8.72%±0.00 (45ms) | 11.07%±0.00 (685ms) | 11.86%±0.00 (4.45s) |
| Random Insertion | 4.46%±0.08 (41ms) | 7.57%±0.08 (575ms) | 9.63%±0.11 (4.34s) |
| Farthest Insertion | 2.38%±0.00 (57ms) | 5.50%±0.00 (909ms) | 7.58%±0.00 (5.68s) |
| LKH3 | 0.00%* (18s) | 0.00%* (5m) | 0.00%* (21m) |

**Travelling salesman problem (TSP).** Table 4 shows the optimality gaps of our approach and its variants on TSP. All of our models were trained using random insertion as the CO heuristic. For PBGE, we sampled 10 solutions from $h(\hat{\boldsymbol{y}} \mid \mathcal{G}, \boldsymbol{s})$ and 10 solutions from $h(\hat{\boldsymbol{y}} \mid \mathcal{G})$.

The decoder used at test time is listed after the name of the respective method in parentheses. "Random ins., 20 runs" refers to running the random insertion algorithm 20 times and using the best tour as the final output. Greedy search starts from an arbitrary node, and follows the edge with the highest score to select the next node. This process of greedily following the best edge is repeated until each node has been visited once. To ensure that the resulting tour is valid, edges that lead to nodes that have already been visited are excluded. Sampling simply refers to sampling multiple solutions and using the best one. "+ 2OPT" refers to improving the decoded solution using 2OPT local search (Croes, 1958).

Figure 4 shows the Pareto frontier of self-supervised methods on TSP-100. PBGE achieves Pareto-optimality w.r.t. our baselines when decoding with 100 random insertion runs.

# 7 Conclusion

We introduced a method to improve existing heuristics for CO using GNNs. The GNN predicts parameters, which are used as input for the non-learned heuristic to produce a high-quality solution to the CO problem. The GNN is trained based on the CO problem's downstream objective, without the need for labeled data. To achieve this, we used gradient estimation to backpropagate through the heuristic. We proposed a novel gradient estimation scheme for this purpose, which we called preference-based gradient estimation (PBGE).

**Limitations and future work.** Incorporating a CO heuristic during training means that the training process is more computationally intensive compared to competing approaches. This also means that an existing probabilistic heuristic is required for our approach, and the choice of this heuristic has a large impact on the final performance. From a theoretical perspective, the derivation of PBGE makes use of approximations, limiting its accuracy.

There are several ways in which this work can be extended. Firstly, generalizing PBGE to nonlinear objectives would greatly broaden its applicability. Secondly, while our current theoretical analysis tackles expressivity, further investigation into learnability could provide a better understanding of PBGE's capabilities. Lastly, we only experimented on CO problems for which solutions can be represented in terms of the graph's edges. More insight could be gained from applying this approach to different kinds of CO problems and to larger CO instances.

## Acknowledgments

The authors would like to thank Aneesh Barthakur, Matteo Palmas, Roman Freiberg, and Jiaqi Wang for fruitful discussions. The authors thank the International Max Planck Research School for Intelligent Systems (IMPRS-IS) for supporting Arman Mielke. The contributions of Thilo Strauss were carried out while employed with ETAS Research.

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

# A  Combinatorial optimization problems and heuristics

## A.1  Minimum $k$-cut problem

### A.1.1  Problem Definition

We are given a connected, undirected graph $\mathcal{G} = (V, E, w)$ with edge weights $w : E \to \mathbb{R}_{>0}$, as well as a desired number of connected components $k \in \mathbb{N}, 2 \le k \le |V|$. The goal is to find a set of edges $C \subseteq E$ with minimal total weight whose removal leaves $k$ connected components. This set is called a *minimum $k$-cut*. Formally, we are optimizing

$$\min_{C \subseteq E} \sum_{e \in C} w(e) \quad \text{such that graph } (V, E \backslash C) \text{ has } k \text{ connected components.}$$

### A.1.2  Karger's algorithm

Karger's algorithm (Karger, 1993) is a Monte Carlo algorithm for the minimum $k$-cut problem.

The algorithm is based on the *contraction operation*: An edge $e = \{x, y\}$ is contracted by merging its nodes $x$ and $y$ into a new node $xy$. For clarity, we will call the node that results from this merger a meta-node. Every edge that was incident to exactly one of the two merged nodes is now altered to instead be incident to the meta-node $xy$: An edge $\{x, z\}$ or $\{y, z\}$ becomes $\{xy, z\}$. This may result in parallel edges, meaning that the resulting graph is a multigraph. All edges $\{x, y\}$ are removed, so that the resulting multigraph contains no self-loops.

Karger's algorithm works by repeatedly sampling an edge, where the probability of each edge is proportional to its weight, then contracting that edge. This is repeated until there are only $k$ nodes left. Each of these remaining $k$ meta-nodes represents a connected component in the original graph, with each node of the original graph that was subsequently been merged into that meta-node belonging to this connected component. Any edge in the original graph that spans between two connected components is cut.

Since this algorithm is not guaranteed to find the minimum $k$-cut, a common strategy is to run the algorithm repeatedly and use the smallest found cut as the final result.

### A.1.3  Karger–Stein algorithm

The Karger–Stein algorithm (Karger & Stein, 1993) is a recursive version of Karger's algorithm, shown in Algorithm 2.

## A.2  Travelling Salesman Problem (TSP)

### A.2.1  Problem Definition

We are given a complete directed or undirected graph $\mathcal{G} = (V, E, w)$ with edge weights $w : E \longrightarrow \mathbb{R}$. Our goal is to find a minimum weight Hamiltonian cycle in $\mathcal{G}$. A Hamiltonian cycle, also called a tour, is a sequence of nodes where each node in the graph appears exactly once, each node in the sequence is adjacent to the previous node, and the first node is adjacent to the last node. The weight of the cycle is the sum of all edge weights of edges between nodes that appear next to each other in the cycle, including the edge from the last to the first node. Instead of representing a tour as a sequence of nodes, we will represent it as the set of edges between nodes that appear next to each other in the sequence. Formally, we are optimizing

$$\min_{T \subseteq E} \sum_{e \in T} w(e) \quad \text{such that } T \text{ forms a Hamiltonian cycle.}$$

The nodes are commonly called cities, and the weight of an edge is commonly called the distance between the two cities. The TSP is NP complete.

---

**Algorithm 2** Karger–Stein algorithm

---

KARGER–STEIN
**Input:** connected, undirected graph $\mathcal{G} = (V, E, w)$
**if** $|V| \leq 6$ **then**
  **return** CONTRACT$(\mathcal{G}, 2)$
**else**
  target $t \leftarrow \left\lceil \dfrac{|V|}{\sqrt{2}} + 1 \right\rceil$
  $\mathcal{G}_1 \leftarrow$ CONTRACT$(\mathcal{G}, t)$
  $\mathcal{G}_2 \leftarrow$ CONTRACT$(\mathcal{G}, t)$
  **return** $\min\{$KARGER–STEIN$(\mathcal{G}_1)$, KARGER–STEIN$(\mathcal{G}_2)\}$ ▷ Return the lower-weight cut
**end if**

CONTRACT
**Input:** connected, undirected graph $\mathcal{G} = (V, E, w)$, target number of nodes $t$
**while** $|V| > t$ **do**
  $\mathcal{G} \leftarrow$ sample edge in $\mathcal{G}$ and contract it
**end while**
**return** $\mathcal{G}$

---

**Metric TSP.** The metric version of TSP additionally assumes that the distances between the cities form a metric. This means that

- the graph is undirected (or $w(x, y) = w(y, x)$ for all $x, y \in V$),

- distances between cities at different locations are positive, and

- the edge weights satisfy the triangle inequality, i.e. $w(x, y) + w(y, z) \geq w(x, z)$ for all $x, y, z \in V$.

**Euclidean TSP.** Euclidean TSP is a special case of metric TSP in which the cities are located at points in the unit square and the distances between the cities are the Euclidean distances between the respective points. Research on solving the TSP using neural networks often focuses on this version of the problem (Kool et al., 2019, Joshi et al., 2019; 2021).

### A.2.2 Random insertion algorithm

The random insertion algorithm (Karg & Thompson, 1964) is a Monte Carlo algorithm for the TSP.

Since a Hamiltonian cycle to a given graph $\mathcal{G} = (V, E)$ is required to contain every $v \in V$ exactly once, it is a straightforward approach to iteratively sample and remove nodes from $V$ until it is empty. The random insertion algorithm, as suggested by Karg and Thompson, begins by selecting two nodes $s, t \in V$ at random and adds the edges $(s, t)$ and $(t, s)$ to an initial cycle. In order to extend the cycle to include all nodes, the algorithm now samples a node $v \in V \setminus \{s, t\}$ and selects the edges $(x, v)$ and $(v, y)$ such that $x$ and $y$ are already part of the partial cycle with $x \neq y$ and such that the sum of the metric distances of $(x, v)$ and $(v, y)$ is minimal.

The cycles obtained in this way are at most $(\lceil \log_2 |V| \rceil + 1)$ times longer than the optimal cycle (Rosenkrantz et al., 1977).

The algorithm is summarized in Algorithm 3.

## B  Proofs

We assume that the edges $E$ of a graph $\mathcal{G} = (V, E, w)$ are in an arbitrary but fixed order. This means that the scores assigned to the edges by a GNN can be represented as a vector $\boldsymbol{s} \in \mathbb{R}^{|E|}$. We use $\boldsymbol{s}_e$ to denote

---

**Algorithm 3** Random insertion

---

    **Input:** connected, undirected graph $\mathcal{G} = (V, E, w)$
    $\mathcal{T} \leftarrow$ TSP tour consisting of one random node
    $v \leftarrow$ sample node in $V$ that is not part of $\mathcal{T}$ yet
    $\mathcal{T} \leftarrow$ insert $v$ into $\mathcal{T}$ to form a loop of two nodes
    **for** $i \in \{1, \ldots, |V|\}$ **do**
        $v \leftarrow$ sample node in $V$ that is not part of $\mathcal{T}$ yet
        $\mathcal{T} \leftarrow$ insert $v$ into $\mathcal{T}$ at the point in the tour $\mathcal{T}$ where it increases the tour's length by the least amount
    **end for**
    **return** $\mathcal{G}$

---

the entry of vector $\boldsymbol{s}$ associated with edge $e \in E$. A subset of edges $\hat{\boldsymbol{y}}$ can be represented as $\hat{\boldsymbol{y}} \in \{0, 1\}^{|E|}$, where a 1 indicates that the respective edge is in the set. For notational simplicity, we will still write $e \in \hat{\boldsymbol{y}}$ for edges that are in this subset.

A probabilistic CO heuristic for an edge subset problem defines a probability distribution over the subsets of edges. We use $h(\hat{\boldsymbol{y}} \mid \mathcal{G}, \boldsymbol{s})$ to denote the probability that the output is edge subset $\hat{\boldsymbol{y}}$ given input graph $\mathcal{G}$ and edge scores $\boldsymbol{s}$, as described in Section 5.

We use $\sigma$ to denote the element-wise sigmoid function $\sigma(x) = \frac{e^x}{1+e^x}$.

### B.1 Probability of finding the minimum $k$-cut using the Karger–Stein algorithm

Let $\mathcal{G} = (V, E)$ be an undirected graph. During the first $i - 1$ iterations of the modified Karger's algorithm, some edges have been merged away. Let $E_i$ be the set of edges that are left at iteration $i$ (this means $E_1 = E$). The probability that a specific edge $e$ is selected for contraction at iteration $i$ is

$$p_i(e) = \frac{1 - \sigma(\boldsymbol{s}_e)}{\sum_{e' \in E_i} \left(1 - \sigma(\boldsymbol{s}_{e'})\right)}.$$

> **Theorem 2.** *Let $\mathcal{G} = (V, E)$ be an undirected graph, and let $\boldsymbol{y} \in \{0, 1\}^{|E|}$ be a minimum $k$-cut on $\mathcal{G}$. Moreover, let $h(\hat{\boldsymbol{y}} \mid \mathcal{G}, \boldsymbol{s})$ be the probability distribution defined by Karger's algorithm guided by parameters $\boldsymbol{s} \in \mathbb{R}^{|E|}$, and let $\sigma$ be the element-wise sigmoid function. Here, $\sigma(\boldsymbol{s}) \rightarrow \boldsymbol{y}$ means that $\sigma(\boldsymbol{s}_e) \rightarrow 1$ if $y_e = 1$ and $\sigma(\boldsymbol{s}_e) \rightarrow 0$ if $y_e = 0$. Then,*
>
> $$\lim_{\sigma(\boldsymbol{s}) \rightarrow \boldsymbol{y}} h(\boldsymbol{y} \mid \mathcal{G}, \boldsymbol{s}) = 1.$$

In other words, in the limit, Karger's algorithm guided by $\boldsymbol{s}$ always finds optimal solution $\boldsymbol{y}$.

*Proof.* Assuming that no edge in $\boldsymbol{y}$ has been contracted yet, the probability that the edge that is selected for contraction in iteration $i$ is in $\boldsymbol{y}$ is

$$p_i(e \in \boldsymbol{y}) = \sum_{e \in \boldsymbol{y}} p_i(e) = \frac{\sum_{e \in \boldsymbol{y}} \left(1 - \sigma(\boldsymbol{s}_e)\right)}{\sum_{e' \in E_i} \left(1 - \sigma(\boldsymbol{s}_{e'})\right)}.$$

Karger's algorithm outputs a given cut $\boldsymbol{y}$ if and only if no edge in $\boldsymbol{y}$ is contracted by the algorithm (see Karger & Stein (1993), lemma 2.1). Let $k$ be the parameter for minimum $k$-cut, i.e. the number of desired connected components. Karger's algorithm will always terminate after $|V| - k$ contraction steps. The probability that no edge in $\boldsymbol{y}$ is contracted during the $|V| - k$ contraction steps is

$$h(\boldsymbol{y} \mid \mathcal{G}, \boldsymbol{s}) = \prod_{i=1}^{|V|-k} \left(1 - p_i(e \in \boldsymbol{y})\right) = \prod_{i=1}^{|V|-k} \left(1 - \frac{\sum_{e \in \boldsymbol{y}} \left(1 - \sigma(\boldsymbol{s}_e)\right)}{\sum_{e' \in E_i} \left(1 - \sigma(\boldsymbol{s}_{e'})\right)}\right).$$

In the numerator, $\lim_{\sigma(\boldsymbol{s}) \to \boldsymbol{y}} \sum_{e \in \boldsymbol{y}} \left(1 - \sigma(\boldsymbol{s}_e)\right) = 0$, since all $\sigma(\boldsymbol{s}_e)$ go to 1. The denominator is greater than zero, because there is at least one edge left that is not in $\boldsymbol{y}$, otherwise the algorithm would be finished. Since the fraction goes towards zero, all terms of the product go towards 1. Formally, each term in the product corresponds to the conditional probability of not contracting any edge in $\boldsymbol{y}$ given previous contractions, but since all factors tend to 1, their finite product does as well. Therefore, we arrive at

$$\lim_{\sigma(\boldsymbol{s}) \to \boldsymbol{y}} h(\boldsymbol{y} \mid \mathcal{G}, \boldsymbol{s}) = 1.$$

This conclusion holds for any minimum $k$-cut $\boldsymbol{y}$, even if multiple optimal cuts exist, since the argument depends only on the edges of the specific $\boldsymbol{y}$. □

The result can trivially be extended to the Karger–Stein algorithm, a variant of Karger's algorithm, and to weighted graphs $\mathcal{G} = (V, E, w)$.

### B.2   Finding the optimal TSP tour using the Christofides algorithm

> **Theorem 3.** *Let $\mathcal{G} = (V, E, w)$ be an undirected metric graph, and let $\boldsymbol{y}$ be an optimal TSP tour on $\mathcal{G}$. Moreover, let $h_{\mathcal{G}}$ be the (deterministic) Christofides algorithm guided by parameters $\boldsymbol{s} \in \mathbb{R}^{|E|}$, and let $\sigma$ be the element-wise sigmoid function. Here, $\sigma(\boldsymbol{s}) \to \boldsymbol{y}$ means that $\sigma(\boldsymbol{s}_e) \to 1$ if $y_e = 1$ and $\sigma(\boldsymbol{s}_e) \to 0$ if $y_e = 0$. Then,*
> $$\lim_{\sigma(\boldsymbol{s}) \to \boldsymbol{y}} h_{\mathcal{G}}(\boldsymbol{s}) = \boldsymbol{y}.$$

*Proof.* Let $\mathcal{G}' = (V, E, w')$ be the graph that the heuristic receives as input after modification through $\boldsymbol{s}$, i.e. $w'(e) = (1 - \sigma(\boldsymbol{s}_e))w(e)$. As $\sigma(\boldsymbol{s})$ approaches $\boldsymbol{y}$, meaning that $\sigma(\boldsymbol{s}_e) \to 1$ if $y_e = 1$ and $\sigma(\boldsymbol{s}_e) \to 0$ if $y_e = 0$, the edge weights of $\mathcal{G}'$ approach

$$\lim_{\sigma(\boldsymbol{s}) \to \boldsymbol{y}} w'(e) = \lim_{\sigma(\boldsymbol{s}) \to \boldsymbol{y}} \left(1 - \sigma(\boldsymbol{s}_e)\right)w(e) = \begin{cases} 0 & \text{if } e \in \boldsymbol{y}, \\ w(e) & \text{if } e \notin \boldsymbol{y}. \end{cases}$$

This means for any $\varepsilon, \delta > 0$, the GNN will reach a $\sigma(\boldsymbol{s})$ such that $w'(e) \geq w(e) - \delta$ for all $e \notin \boldsymbol{y}$ and $w'(\boldsymbol{y}) := \sum_{e \in \boldsymbol{y}} w'(e) \leq \varepsilon$.

The Christofides algorithm begins by calculating a minimum spanning tree $T_{\min}$. By choosing $\varepsilon = \delta$, $2\varepsilon < \min_{e \in E} w(e)$, we can guarantee that, once the respective $\sigma(\boldsymbol{s})$ is reached, the minimum spanning tree looks as follows. $T_{\min}$ consists of all edges in tour $\boldsymbol{y}$, except for the edge with the highest modified weight in $\boldsymbol{y}$:

$$T_{\min} = \boldsymbol{y} \setminus \{e_r\}, \qquad \text{where } e_r = \arg\max_{e \in \boldsymbol{y}} w'(e).$$

(There may be more then one highest-weight edge in $\boldsymbol{y}$, in which case the minimum spanning tree is not unique. However, all minimum spanning trees are of this form.)

$T_{\min}$ is a spanning tree, since it is a path that traverses all nodes. To see why $T_{\min}$ is a minimum spanning tree, consider any set of edges $T \subseteq E$.

- If $T$ contains any edge $e \notin \boldsymbol{y}$, then

$$w'(T) \geq w'(e) \geq w(e) - \delta > \varepsilon \geq w'(\boldsymbol{y}) > w'(T_{\min}).$$

  Therefore, $S$ cannot be a minimum spanning tree.

- If $T = \boldsymbol{y}$, then $T$ is a cycle and therefore not a tree.

- If $T = \boldsymbol{y} \setminus \{e\}$ for any edge $e \in \boldsymbol{y}$, then $w'(e) \leq w'(e_r)$ and therefore

$$w'(T) = w'(T_{\min}) - w'(e) + w'(e_r) \geq w'(T_{\min}).$$

  $T_{\min}$ is still a minimum spanning tree. Note that the case $w'(T) = w'(T_{\min})$ only occurs if $T = T_{\min}$ or if there are multiple edges in $\boldsymbol{y}$ with the highest modified weight, in which case both $T$ and $T_{\min}$ are minimum spanning trees of the form described above.

- If $T$ includes only edges in $\boldsymbol{y}$, but is missing two or more edges from $\boldsymbol{y}$, then it is not a spanning tree. $T$ is either not connected, and therefore not a tree, or it does not cover all nodes.

The Christofides algorithm picks (one of) the minimum spanning tree(s) of the form described above. Next, it calculates the set of nodes in this minimum spanning tree with odd degree. Since $T_{\min}$ is a path, the only nodes with odd degree are the endpoints, which are the two nodes connected by $e_r$.

Then, the algorithm forms a minimum-weight perfect matching on this set of odd-degree nodes. Since the set only contains two nodes, the matching is unique and consists of the single edge $e_r$.

In the next step, the Christofides algorithm combines this matching with the minimum spanning tree $T_{\min}$. In other words, it adds edge $e_r$ to $T_{\min}$, resulting in $T_{\min} \cup \{e_r\} = \boldsymbol{y}$.

Finally, the Christofides algorithm modifies the tour such that each node is only visited once. However, since $\boldsymbol{y}$ is already a Hamiltonian cycle, no changes are made and $\boldsymbol{y}$ is returned. $\qquad\square$

### B.3 Probability of finding the optimal TSP tour using an insertion algorithm

> **Theorem 4.** *Let $\mathcal{G} = (V, E, w)$ be an undirected metric graph, and let $\boldsymbol{y}$ be an optimal TSP tour on $\mathcal{G}$. Moreover, let $h(\hat{\boldsymbol{y}} \mid \mathcal{G}, \boldsymbol{s})$ be the probability distribution defined by an insertion algorithm for the TSP (e.g. random insertion), guided by parameters $\boldsymbol{s} \in \mathbb{R}^{|E|}$. There exists a set of parameters $\boldsymbol{s} \in \mathbb{R}^{|E|}$ such that*
> $$h(\boldsymbol{y} \mid \mathcal{G}, \boldsymbol{s}) = 1.$$

In other words, there are GNN output parameters that *guarantee* that the optimal TSP tour is found, even if a randomized insertion algorithm such as random insertion is used.

*Proof.* Let $v_1, \ldots, v_n$ be the order of the nodes as they are traversed by the optimal tour $\boldsymbol{y}$, starting at an arbitrary starting node $v_0$ and going in an arbitrary direction. We assign a 2D position $f(v_i) = (\alpha i, \varepsilon i^2)$ to each node $v_i$. We choose $\varepsilon$ so small that $\varepsilon n^2 \ll \alpha$. This puts the nodes in a line that is almost straight and only slightly curves upwards based on $\varepsilon i^2$. Let $d(v_i, v_j)$ be the distance between nodes $v_i, v_j$ in this 2D space.

Let $\mathcal{G}' = (V, E, w')$ be the graph that the heuristic receives as input after modification through $\boldsymbol{s}$, i.e. $w'(e) = (1 - \sigma(\boldsymbol{s}_e))w(e)$. We will assign the scores $\boldsymbol{s}$ such that for each edge $e = \{u, v\}$, the modified edge weight is $w'(e) = d(u, v)$. Since $0 < \sigma(\boldsymbol{s}_e) < 1$, the edge weights can only be scaled down. We will choose $\alpha, \varepsilon$ so small that the smallest edge weight in $w$ is larger than the largest distance in $d$. Since the $x$ and $y$ coordinates of each node $v_i$ increase monotonically with increasing $i$, the largest distance in $d$ is between the first and last node, $d(v_1, v_n)$. We choose $\alpha, \varepsilon$ small enough that $\min_{e \in E} w(e) > d(v_1, v_n)$. This means that for each edge $e = \{u, v\}$, $w(e) > d(u, v)$, and therefore there exists a score $\boldsymbol{s}_e$ such that $(1 - \sigma(\boldsymbol{s}_e))w(e) = d(u, v)$.

We have established that there exists a set of scores $\boldsymbol{s}$ such that the modified graph $\mathcal{G}'$ is a metric graph with metric $d$. It remains to show that on this modified graph, an insertion algorithm will always find TSP tour $\boldsymbol{y}$. We will prove by induction that the insertion algorithm always maintains a monotone tour, i.e. a tour that traverses its nodes in the same order as $\boldsymbol{y}$ (up to reversal). This means that after the last insertion step, the algorithm returns $\boldsymbol{y}$.

The first three nodes that are chosen are always connected by a minimum-length cycle, which is necessarily monotone (up to reversal). For each node that is inserted into the tour afterwards, the tour remains monotone

regardless of how the node is selected. Suppose a current tour visits a subset $S \subset V$ in monotone order. Let $v_k \in V \setminus S$ be the next node selected by the insertion algorithm, and let $p$ (resp. $q$) be the largest index in $S$ with $p < k$ (resp. the smallest with $q > k$), when such indices exist. If $p$ and $q$ exist, i.e. if the tour already contains nodes on either side of $v_k$ in the ordering $v_1, \ldots, v_n$, then inserting $v_k$ at an edge $\{a, b\}$ changes the length by

$$\Delta(a, b; v_k) = d(a, v_k) + d(v_k, b) - d(a, b).$$

Since $\varepsilon$ is negligibly small compared to $\alpha$, for indices $i \leq k \leq j$ one has $d(v_i, v_k) + d(v_k, v_j) - d(v_i, v_j) < \alpha$. In contrast, if $k < i \leq j$ (or $i \leq j < k$) the increase is strictly larger than $\alpha$. Because the current tour is monotone, the unique edge whose endpoints straddle $k$ in the path order is $\{v_p, v_q\}$; $v_k$ is inserted there because this insertion yields $\Delta(v_p, v_q; v_k) < \alpha$, while every other insertion yields $\Delta > \alpha$.

If only one of $p$ or $q$ exists (i.e. $v_k$ lies outside the current hull of $S$), the unique best insertion is adjacent to that boundary node by the same calculation. Thus the minimum-increase insertion places $v_k$ between this boundary node and either of its neighbors in the current tour. Note that because the tour is monotonous, one of these neighbors is the boundary node at the other end of $S$. The $\varepsilon$-perturbation ensures that out of the two options, the insertion algorithm favors inserting the new node between the first and last node in $S$. This preserves monotonicity as the newly inserted node becomes the new first node. To see why this option is always taken, let $v_f, v_s, v_l$ be the first, second, and last node in $S$ respectively, and let $k < f$ (this is without loss of generality, as the case $k > l$ is the same with inverted order of the nodes). As discussed above, the choice is between inserting $v_k$ at edge $\{v_f, v_s\}$ (breaking monotonicity) or at edge $\{v_f, v_l\}$ (preserving monotonicity). The increase incurred by each respective choice is

$$\Delta(v_f, v_s; v_k) = d(v_f, v_k) + d(v_k, v_s) - d(v_f, v_s),$$
$$\Delta(v_f, v_l; v_k) = d(v_f, v_k) + d(v_k, v_l) - d(v_f, v_l).$$

The points $v_k, v_f, v_s, v_l$ form a (strictly) convex quadrilateral. It is easy to see that in a convex quadrilateral, the sum of the lengths of the diagonals is greater than the sum of the lengths of two opposing sides.[3] We therefore have $d(v_k, v_s) + d(v_f, f_l) > d(v_k, v_l) + d(v_f, v_s)$ and therefore $\Delta(v_f, v_s; v_k) > \Delta(v_f, v_l; v_k)$. Since inserting $v_k$ at edge $\{v_f, v_l\}$ incurs a smaller increase in tour length than $\{v_f, v_s\}$, the insertion algorithm inserts the node at edge $\{v_f, v_l\}$ and monotonicity is preserved.

When all nodes have been inserted, the tour is exactly $v_1, \ldots, v_n$ or its reversal, i.e. the same tour as $\boldsymbol{y}$, independently of how the next node to insert is selected. This proves the claim. $\square$

## C Implementation details

### C.1 Residual gated graph convnets

We use residual gated graph convnets (Bresson & Laurent, 2017), but adapt them to include edge features $\boldsymbol{e}_{ij}^l$ and a dense attention map $\boldsymbol{\eta}_{ij}^l$ following Joshi et al. (2019). The input node features $\boldsymbol{x}_i^0$ and edge features $\boldsymbol{e}_{ij}^0$ are first pre-processed using a single-layer MLP (multi-layer perceptron) for each of the two. Each further layer is computed as follows:

$$\boldsymbol{x}_i^{l+1} = \boldsymbol{x}_i^l + \mathrm{ReLU}\left(\mathrm{BN}\left(W_1^l \boldsymbol{x}_i^l + \sum_{j \in \mathcal{N}_i} \boldsymbol{\eta}_{ij}^l \odot W_2^l \boldsymbol{x}_j^l\right)\right) \text{ with } \boldsymbol{\eta}_{ij}^l = \frac{\sigma(\boldsymbol{e}_{ij}^l)}{\sum_{j' \in \mathcal{N}_i} \sigma(\boldsymbol{e}_{ij'}^l + \varepsilon)} \in \mathbb{R}^d,$$

$$\boldsymbol{e}_{ij}^{l+1} = \boldsymbol{e}_{ij}^l + \mathrm{ReLU}\left(\mathrm{BN}\left(W_3^l \boldsymbol{e}_{ij}^l + W_4^l \boldsymbol{x}_i^l + W_5^l \boldsymbol{x}_j^l\right)\right),$$

where $W_1^l, \ldots, W_5^l \in \mathbb{R}^{d \times d}$ are learnable weights, $d$ is the hidden dimension, ReLU is the rectified linear unit, BN is batch normalization, $\sigma = \frac{e^x}{1+e^x}$ is the element-wise sigmoid function, and $\varepsilon$ is an arbitrary small value. $\odot$ denotes the Hadamard product, and $\mathcal{N}_i$ denotes the set of nodes that are adjacent to $i$.

---

[3]Let $a, b, c, d$ be the points of a (strictly) convex quadrilateral. We want to show that $\|a - d\| + \|b - c\| > \|a - b\| + \|c - d\|$. Let $x$ be the point in which the diagonals intersect. We can therefore divide the diagonals into $\|a - d\| + \|b - c\| = \|x - a\| + \|x - b\| + \|x - c\| + \|x - d\|$. By the triangle inequality, we have $\|x - a\| + \|x - b\| > \|a - b\|$ and $\|x - c\| + \|x - d\| > \|c - d\|$. These are strict inequalities since $x, a, b$ (resp. $x, c, d$) are not collinear. Hence we arrive at $\|x - a\| + \|x - b\| + \|x - c\| + \|x - d\| > \|a - b\| + \|c - d\|$.

The final edge-level output is calculated from the last layer's edge features $\boldsymbol{e}_{ij}^l$ using another MLP. $f(\mathcal{G}) \in \mathbb{R}^{|E|}$ refers to applying this GNN on a graph $\mathcal{G}$.

### C.2 Decoding using the Karger–Stein algorithm

In the case of the Karger–Stein algorithm, we noticed empirically that simply modifying the input graph can lead to degenerate behavior during testing. The Karger–Stein algorithm uses the graph's edge weights in two places: (1) when sampling an edge for contraction and (2) when comparing the cuts that resulted from different recursion arms. We noticed that the performance of our overall method can be improved when using a model trained with the setting described in Section 5.2 by using the modified edge weights for the first case and the original edge weights for the second case. Intuitively, if the GNN makes a mistake when scaling the edge weights, using the original edge weights for comparing cuts can allow the Karger–Stein algorithm to find the optimal cut regardless.

## D   Details regarding baselines

### D.1   Supervised training with binary cross entropy loss

The task is treated as an edge-level binary classification task. The network is trained using a BCE loss:

$$\hat{y} = \sigma\big(f_\theta(\mathcal{G})\big)$$

$$\mathcal{L}_{\text{supervised BCE}}(\mathcal{G}, y) = \text{BCE}(\hat{y}, y) = \sum_{i=1}^{|E|} y_i \log \hat{y}_i + (1 - y_i) \log(1 - \hat{y}_i)$$

where $f_\theta$ is a GNN, $\mathcal{G}$ is the input graph, $y \in \{0,1\}^{|E|}$ is the ground truth solution and $\hat{y} \in (0,1)^{|E|}$ is the predicted solution.

A ground truth label of 1 represents that an edge belongs to a minimum $k$-cut or a TSP tour.

### D.2   REINFORCE

Recall that the REINFORCE algorithm (Williams, 1992), also known as the score function estimator, calculates

$$\nabla_\theta \mathbb{E}_{y \sim p_\theta(x)}\big[J(y)\big] = \mathbb{E}_{y \sim p_\theta(x)}\big[J(y)\nabla_\theta \log p_\theta(x)\big]$$

where $J$ is an objective function, and $p_\theta(x)$ is a probability distribution parameterized by $\theta$.

We assume that $p_\theta$ is a discrete constrained exponential family distribution, i.e.

$$p_\theta(x) = \begin{cases} \dfrac{\exp\big(\langle x, \theta \rangle\big)}{\sum_{x'} \exp\big(\langle x', \theta \rangle\big)} & \text{if } x \text{ satisfies the constraints} \\ 0 & \text{otherwise} \end{cases}$$

For valid $x$,

$$\log p_\theta(x) = \langle x, \theta \rangle - A(\theta),$$

where $A(\theta)$ is the log-partition function

$$A(\theta) = \log\left(\sum_{x' \in \mathcal{C}} \exp\big(\langle x', \theta \rangle\big)\right).$$

Since $\nabla_\theta A(\theta) = \mathbb{E}_{y \sim p_\theta(x)}[y]$, we get

$$\nabla_\theta \log p_\theta(x) = x - \mathbb{E}_{y \sim p_\theta(x)}[y].$$

Inserting this into the REINFORCE formula gives us

$$\nabla_\theta \mathbb{E}_{y \sim p_\theta(x)}\big[J(y)\big] = \mathbb{E}_{y \sim p_\theta(x)}\left[J(y)\Big(y - \mathbb{E}_{y' \sim p_\theta(x)}[y']\Big)\right].$$

Using the Gumbel-max trick, we sample from $p_\theta$ by sampling $\varepsilon \sim \mathrm{Gumbel}(0,1)$, then calculating $y \coloneqq h(\theta + \varepsilon)$.

Estimating the outer expectation by sampling once and the inner expectation by sampling $N$ times, we arrive at Algorithm 4.

---

**Algorithm 4** REINFORCE

---

**Input:** distribution parameter $\theta$
$\varepsilon \sim \mathrm{Gumbel}(0,1)$
$y \leftarrow h(\theta + \varepsilon)$

$$\nabla_\theta J(y) \leftarrow J(y)\left(y - \frac{1}{N}\sum_{i=1}^{N} y_i\right)$$
$\quad\quad\quad$ where $\varepsilon_i \sim \mathrm{Gumbel}(0,1)$
$\quad\quad\quad\quad\quad y_i \leftarrow h(\theta + \varepsilon_i)$
**return** $\nabla_\theta J(y)$

---

### D.3   Implicit maximum likelihood estimator (I-MLE)

I-MLE (Niepert et al., 2021) allows estimating gradients with respect to the parameters of discrete exponential family distributions. This can be used to backpropagate through CO solvers as follows. In the forward pass, perturb the input $\boldsymbol{\theta} \in \mathbb{R}^n$ to the CO solver using noise $\boldsymbol{\epsilon} \sim \rho(\boldsymbol{\epsilon})$ sampled from a suitable noise distribution. Then run the CO solver on the perturbed input $\boldsymbol{\theta} + \boldsymbol{\epsilon}$, obtaining output $\boldsymbol{z}$.

In the backward pass, assume we know the gradient of the loss w.r.t. to $\boldsymbol{z}$, $\nabla_{\boldsymbol{z}}\mathcal{L}$. First, obtain a modified input $\boldsymbol{\theta}'$ for which we can expect better outputs compared to $\boldsymbol{\theta}$. One generally applicable option suggested by Niepert et al. (2021) is $\boldsymbol{\theta}' = \boldsymbol{\theta} - \lambda \nabla_{\boldsymbol{z}}\mathcal{L}$, where $\lambda$ is a hyperparameter. Using the same noise $\boldsymbol{\epsilon}$ as in the forward pass, perturb $\boldsymbol{\theta}'$ and run the CO solver on $\boldsymbol{\theta}' + \boldsymbol{\epsilon}$, obtaining $\boldsymbol{z}'$. Finally, return the estimated gradient $\nabla_{\boldsymbol{\theta}}\mathcal{L} \approx \boldsymbol{z} - \boldsymbol{z}'$.

This produces biased gradient estimates, but with much smaller variance than REINFORCE. To further reduce variance, this procedure can be repeated $S$ times, sampling new noise $\boldsymbol{\epsilon}_i \sim \rho(\boldsymbol{\epsilon}_i)$ each time and averaging the results.

We used three noise samples from the Sum-of-Gamma noise distribution suggested by Niepert et al. (2021), using $\kappa = 5$ and 100 iterations.

### D.3.1   Supervised training with I-MLE

The outputs of the GNN are used to guide a CO heuristic. This heuristic outputs a solution to the CO problem, which can be compared to the ground truth solution using a Hamming loss. During backpropagation, I-MLE (Niepert et al., 2021) is used to estimate the gradient of the loss with respect to the GNN's output. This setting has no practical benefit over the simple supervised training using a BCE loss, but it serves to measure the effectiveness of I-MLE.

The training procedure works as follows.

$$s = \sigma\big(f_\theta(\mathcal{G})\big)$$
$$\hat{y} = h(\mathcal{G}, 1 - s)$$
$$\mathcal{L}_{\text{supervised I-MLE}}(\mathcal{G}, y) = \frac{1}{|E|}\sum_{i=1}^{|E|} \hat{y}_i(1 - y_i) + (1 - \hat{y}_i)y_i$$

Note that here, $\hat{y} \in \{0, 1\}^{|E|}$ is guaranteed to be a valid solution to the given CO problem. I-MLE is used to estimate $\frac{\mathrm{d}\mathcal{L}}{\mathrm{d}(1-s)}$.

### D.3.2 I-MLE target distribution

We're using a custom target distribution for I-MLE in the supervised setting. This target distribution is similar to the target distribution for CO problems presented in (Niepert et al., 2021). The idea behind it is to recover the ground truth label from the loss, which is possible when using the Hamming loss:

$$\ell(\hat{y}, y) = \hat{y}(1 - y) + (1 - \hat{y})y$$

$$\frac{\mathrm{d}}{\mathrm{d}\hat{y}}\ell(\hat{y}, y) = 1 - 2y$$

$$y = \frac{1 - \frac{\mathrm{d}}{\mathrm{d}\hat{y}}\ell(\hat{y}, y)}{2}$$

The best value for $\theta$ is $1 - y$ (we have to invert it because the input to the heuristic is inverted).[4] With this we arrive at the following target distribution:

$$\theta' = 1 - y = \frac{1 + \frac{\mathrm{d}}{\mathrm{d}\hat{y}}\mathcal{L}}{2}$$

### D.3.3 Self-supervised training with I-MLE

Since we're using a CO heuristic that guarantees that its outputs are valid solutions to the CO problem, we can use the CO problem's objective function as a loss directly instead of the supervised Hamming loss. The ground truth labels are therefore no longer required.

In the case of minimum $k$-cut, the size of the cut is used as loss function. For TSP, the length of the tour is used.

$$s = \sigma\big(f_\theta(\mathcal{G})\big)$$
$$\hat{y} = h(\mathcal{G}, 1 - s)$$
$$\mathcal{L}_{\text{self-supervised I-MLE}}(\mathcal{G}) = J_{\text{CO}}(\hat{y})$$

where $J_{\text{CO}}$ is the objective function of the CO problem. Note that the CO problem's constraints do not explicitly appear here, because the CO heuristic already guarantees that the constraints are met.

As before, I-MLE is used to estimate $\frac{\mathrm{d}\mathcal{L}}{\mathrm{d}(s-1)}$. In this setting, the general-purpose target distribution for I-MLE presented in (Niepert et al., 2021) is used, setting $\lambda = 20$ as suggested by that paper.

## E   Graph generation

### E.1   Minimum $k$-cut

Many commonly used graph generators create graphs with low-degree nodes. These graphs contain trivial solutions to the minimum $k$-cut problem in which $k - 1$ connected components only contain one node, and one connected component contains all of the remaining nodes. When creating a dataset for minimum $k$-cut, care must therefore be taken to avoid graphs with low-degree nodes.

---

[4]Using $\theta$ in the same sense as Niepert et al. (2021). In our case, $\theta = 1 - s$

**Graphs without edge weights.** A simple method to generate graphs with meaningful solutions to the minimum $k$-cut problem is as follows. Create $k$ fully connected subgraphs of random sizes within a given range. Then, add a random number of edges between random nodes of different subgraphs while ensuring that the resulting graph is connected. An additional benefit of this method is that, if the number of edges added between subgraphs is smaller than the number of nodes in the smallest subgraph by at least two, then the minimum $k$-cut is known from the construction: the cut consists of exactly the edges that were added between subgraphs. However, since all graphs generated this way consist of fully connected subgraphs, these problem instances are limited in diversity.

The range of possible problem instances can be improved by generating graphs of varying density. Start by assigning nodes to $k$ subgraphs of random sizes within a given range. Then, add a random number of edges that connect nodes of different subgraphs. For each subgraph, add edges between random nodes within the same subgraph until all nodes have a higher degree than the number of edges between subgraphs. As long as there are enough edges between subgraphs, the minimum $k$-cut very likely consists of the edges between subgraphs. The minimum node degree and hence the density of the graph depends on the number of edges between subgraphs and therefore on the size of the minimum $k$-cut.

**Graphs with edge weights.** For minimum $k$-cut graphs with edge weights, a graph generator commonly called NOIgen (Nagamochi et al., 1994) (named after the initials of the authors) is often used. NOIgen works by first creating a specified number of nodes and adding edges between random nodes until a specified density is reached (sometimes, a Hamilton path is created first to ensure that the graph is connected). The weights of the edges are chosen uniformly at random. Finally, the nodes are randomly divided into $k$ subgraphs. The weights of edges that connect nodes of different subgraphs are scaled down by a fixed factor.

When testing traditional, non-learned algorithms, the scaling factor is sometimes chosen to be very small, such that the minimum $k$-cut is very likely to consist of the edges between subgraphs (Chekuri et al., 1997). However, this makes the problem trivially easy for GNNs, which can learn that a very low edge weight corresponds to an edge belonging to the minimum $k$-cut. This allows the GNN to disregard the graph structure and therefore circumvent the challenging part of the problem. On the other hand, if the weights of edges between subgraphs are not scaled down enough, the generated graph might have a trivial solution that simply cuts out $k - 1$ nodes.

To combat this problem, we modify NOIgen by controlling not just the weights of edges between subgraphs, but also the number of edges between subgraphs. We add a parameter that specifies which fraction of edges is generated between subgraphs (as opposed to within the same subgraph). Ensuring that there are few enough edges between subgraphs allows for a milder downscaling of their edge weights without introducing a trivial solution. This in turn prevents the GNN from inferring whether an edge belongs to the minimum $k$-cut simply from its weight.

The minimum $k$-cut in these graphs usually consists of the edges between subgraphs, but this is not guaranteed. The ground truth solution is therefore calculated separately to make sure that it reflects the optimal cut, as described in Section E.3. Another benefit of calculating them separately is that we can set the number of subgraphs to a different number than $k$, generating more interesting graphs.

### E.2 Travelling salesman problem

Instances of Euclidean TSP are commonly generated with this simple algorithm:

1. Create a fully connected graph with $n$ nodes

2. For each node, draw a position in the unit square uniformly at random and assign it as node features

3. Calculate the distances between the nodes and assign them as edge features

### E.3 Calculating ground truth labels for supervised training

In general, ground truth labels are generated using a traditional (i.e. non-learned) algorithm. In some settings, the graph can be constructed such that the ground truth solution can be obtained simultaneously from the same construction process, in which case running the traditional algorithm is not necessary.

- Minimum k-cut: For graphs without edge weights, the graphs can be constructed with known ground truth solutions. See Section E.1 for details. For graphs with edge weights, the Karger–Stein algorithm (Karger & Stein, 1993) is run 100 times, and the smallest cut found is treated as the ground truth minimum cut.

- TSP: The well-established Concorde solver (Applegate et al., 2006), which guarantees optimal solutions, is used to generate ground truth labels.

## F  Extended experiments

### F.1  Minimum $k$-cut with $k > 2$

It is common to run the Karger–Stein algorithm repeatedly until the minimum $k$-cut is found with sufficiently large probability. Figure 5 shows for how many graphs the minimum $k$-cut is found in a set number of runs, for $k = 4$ and $k = 3$. For both values of $k$, our approach reduces the number of Karger–Stein runs required to find the minimum $k$-cut with 99% probability by more than half, thus providing a considerable speed up.

### F.2  Extended comparison for TSP

Table 5 compares PBGE with more baselines, including ones that make use of supervised learning.

We also compare against a simple self-supervised baseline that runs random insertion on the input graph 20 times and treats the best solution found as ground truth for a BCE loss. We call this baseline "Best-of-20".

The beam search decoder works similarly to the greedy decoder. It starts with an arbitrary node, then explores the $b$ edges with the highest scores. This gives us $b$ partial solutions. In each iteration, each partial solution is expanded at its last node, and out of the resulting paths, the $b$ best partial solutions are kept. Edges that would lead to invalid tours are ignored. The parameter $b$ is called the *beam width*, and beam search with $b = 1$ corresponds to greedy search.

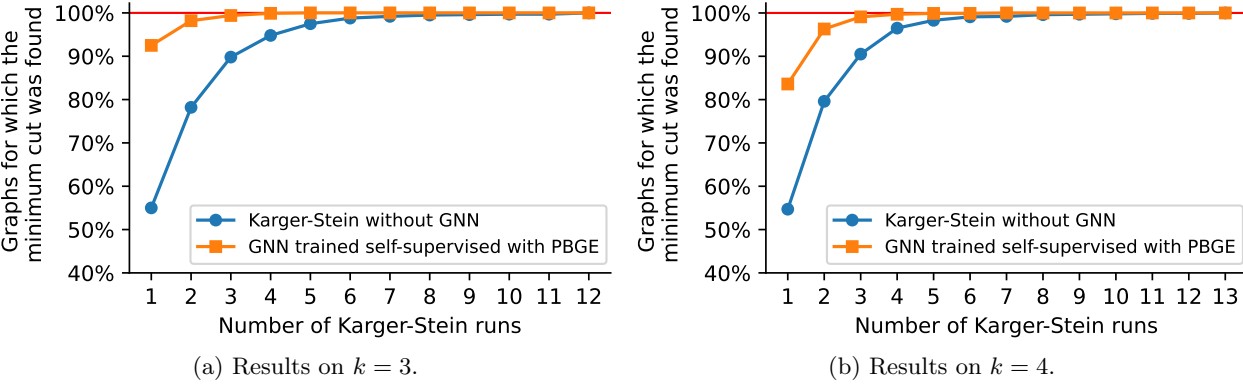

(a) Results on $k = 3$.  (b) Results on $k = 4$.

Figure 5: The number of graphs for which the minimum $k$-cut was found after a given number of Karger–Stein runs, on $k = 3$ and $k = 4$.

Table 5: TSP optimality gaps with mean ± standard deviation calculated over ten evaluation runs on the same model parameters. We group methods by their original paper and indicate the decoder used in parentheses. Results marked with * are values obtained from the indicated papers and therefore do not include standard deviations.

| **Method** (decoder in parentheses) | $n = 20$ | $n = 50$ | $n = 100$ |
|---|---|---|---|
| **Self-supervised** | | | |
| Bello et al. (2017) | | | |
|   (greedy) | 1.42%* | 4.46%* | 6.90%* |
| Deudon et al. (2018) | | | |
|   (greedy) | 0.66%* (2m) | 3.98%* (5m) | 8.41%* (8m) |
|   (greedy + 2OPT) | 0.42%* (4m) | 2.77%* (26m) | 5.21%* (3h) |
|   (sampling 1280 times) | 0.11%* (5m) | 1.28%* (17m) | 12.70%* (56m) |
|   (sampling 1280 times + 2OPT) | 0.09%* (6m) | 1.00%* (32m) | 4.64%* (5h) |
| Khalil et al. (2017) | | | |
|   (greedy) | 1.42%* | 5.16%* | 7.03%* |
| Kool et al. (2019) | | | |
|   (greedy) | 0.34%* | 1.76%* (2s) | 4.53%* (6s) |
|   (sampling 1280 times) | 0.08%* (5m) | 0.52%* (24m) | 2.26%* (1h) |
| REINFORCE (random ins., 20 runs) | 7.98%±0.08 (769ms) | 23.84%±0.10 (11.62s) | 52.83%±0.47 (1.38m) |
| REINFORCE (random ins., 100 runs) | 4.78%±0.03 (3.64s) | 11.24%±0.08 (57.85s) | 31.06%±0.25 (7.53m) |
| I-MLE (random ins., 20 runs) | 10.54%±0.15 (761ms) | 35.87%±0.34 (11.14s) | 61.42%±0.23 (1.37m) |
| I-MLE (random ins., 100 runs) | 6.55%±0.10 (3.68s) | 19.40%±0.24 (56.63s) | 44.42%±0.14 (7.51m) |
| Id. (random ins., 20 runs) | 8.92%±0.13 (762ms) | 31.87%±0.32 (11.10s) | 56.55%±0.33 (1.41m) |
| Id. (random ins., 100 runs) | 5.16%±0.09 (3.66s) | 17.85%±0.09 (55.92s) | 39.69%±0.28 (7.48m) |
| Best-of-20 (random insertion, 20 runs) | 0.40%±0.03 (787ms) | 11.94%±0.11 (11.03s) | 15.82%±0.09 (1.47m) |
| Best-of-20 (random insertion, 100 runs) | 0.10%±0.01 (3.70s) | 5.21%±0.15 (52.82s) | 9.82%±0.09 (7.38m) |
| PBGE **(ours)** (random ins., 20 runs) | 0.18%±0.01 (763ms) | 2.37%±0.02 (11.11s) | 5.12%±0.06 (1.43m) |
| PBGE **(ours)** (random ins., 100 runs) | 0.05%±0.01 (3.73s) | 1.13%±0.03 (53.98s) | 3.67%±0.05 (7.44m) |
| **Supervised**, not directly comparable | | | |
| Joshi et al. (2019) | | | |
|   (greedy) | 0.60%* (6s) | 3.10%* (55s) | 8.38%* (6m) |
|   (beam search, beam width 1280) | 0.10%* (20s) | 0.26%* (2m) | 2.11%* (10m) |
|   (beam search, width 1280 + heuristic) | 0.01%* (12m) | 0.01%* (18m) | 1.39%* (40m) |
| Sun & Yang (2023) | | | |
|   (greedy) | | 0.10%* | 0.24%* |
|   (sampling 16 times) | | 0.00%* | 0.00%* |
| BCE loss (random insertion, 20 runs) | 0.15%±0.01 (787ms) | 0.95%±0.03 (10.92s) | 2.86%±0.04 (1.47m) |
| BCE loss (random insertion, 100 runs) | 0.04%±0.00 (3.75s) | 0.59%±0.02 (54.79s) | 1.75%±0.03 (7.35m) |
| I-MLE (random ins., 20 runs) | 2.67%±0.06 (765ms) | 14.71%±0.18 (11.24s) | 26.57%±0.17 (1.41m) |
| I-MLE (random ins., 100 runs) | 1.17%±0.04 (3.63s) | 9.66%±0.10 (54.80s) | 18.91%±0.11 (7.49m) |
| **Non-learned heuristics** | | | |
| Christofides | 8.72%±0.00 (45ms) | 11.07%±0.00 (685ms) | 11.86%±0.00 (4.45s) |
| Random Insertion | 4.46%±0.08 (41ms) | 7.57%±0.08 (575ms) | 9.63%±0.11 (4.34s) |
| Farthest Insertion | 2.38%±0.00 (57ms) | 5.50%±0.00 (909ms) | 7.58%±0.00 (5.68s) |
| LKH3 | 0.00%* (18s) | 0.00%* (5m) | 0.00%* (21m) |
| **Exact solvers**, not directly comparable | | | |
| Concorde (Applegate et al., 2006) | 0.00%* (1m) | 0.00%* (2m) | 0.00%* (3m) |
| Gurobi | 0.00%* (7s) | 0.00%* (2m) | 0.00%* (17m) |

Table 6: Hyperparameters used for minimum $k$-cut.

| Hyperparameter | Value |
| --- | --- |
| GNN layers | 4 |
| GNN hidden channels | 32 |
| MLP prediction head layers | 2 |
| Optimizer | AdamW |
| Weight Decay | 0.01 |
| Learning rate scheduler | ReduceLROnPlateau |
| Initial learning rate | 0.001 |
| Learning rate scheduler patience | 4 |
| Batch size | 64 graphs |

Table 7: Hyperparameters used for TSP.

| Hyperparameter | Value |
| --- | --- |
| GNN layers | 12 |
| GNN hidden channels | 100 |
| MLP prediction head layers | 3 |
| Optimizer | AdamW |
| Weight Decay | 0.01 |
| Learning rate scheduler | ReduceLROnPlateau |
| Initial learning rate | 0.0001 |
| Learning rate scheduler patience | 4 |
| Batch size | 64 graphs |

## G  Hyperparameters and other details

Table 6 and Table 7 detail the hyperparameters used for experiments on minimum $k$-cut and TSP, respectively.

Our training sets for minimum $k$-cut contain 10,000 graphs, and the validation sets contain 1,000 graphs. For TSP, our training sets consist of 1,000,000 graphs, and our validation sets contain 1,000 graphs.

## H  Hardware resources used for experiments

We used an AMD EPYC 7313 16-Core processor and an NVIDIA H100 GPU in our experiments. Runtime measurements were taken on an 11th Gen Intel Core i7-11850H processor with a base speed of 2.50 GHz, using a single CPU core and no GPU, with a batch size of 1.

