# OpenReview forum: "Preference-Based Gradient Estimation for ML-Guided Approximate Combinatorial Optimization"
_TMLR — Accepted by TMLR_

### Review · Reviewer_cbsU · 2025-11-27

**Summary Of Contributions:**

Summary:

The paper proposes a learning-based framework to enhance existing non-learned heuristics for CO problems, specifically validating the method on the TSP and the Minimum k-cut problem. The core approach involves parameterizing standard heuristics (like Karger-Stein or Random Insertion) by training a GNN to predict edge scores, which are then used to modify the edge weights of the input graph before the heuristic is applied. To enable end-to-end self-supervised training without ground-truth labels, the authors introduce a novel gradient estimation scheme called preference-based gradient estimation (PBGE). PBGE treats the heuristic as a black box and estimates gradients by comparing pairs of solutions sampled from the heuristic, using a proxy distribution to approximate the heuristic's behavior for differentiability. Empirical results demonstrate that the approach improves the performance of the base heuristics, notably on the minimum k-cut problem.


Strengths:

1) Hybrid approach ensuring feasibility: By using the GNN to predict parameters (edge weights) for an off-the-shelf heuristic rather than predicting the solution directly, the method ensures valid solutions are produced without complex constraint management.

2) Self-supervised learning: The framework is fully self-supervised, eliminating the need for expensive pre-computed optimal solutions as ground-truth labels, which is often a bottleneck in CO.

3) Strong empirical results on k-cut: The method demonstrates significant performance gains on the Minimum k-cut problem, improving the optimality gap by an order of magnitude compared to the unaugmented heuristic.

Weaknesses:

1) Unprincipled derivation and approximations: The mathematical derivation of the gradient estimator relies on significant approximations that are not rigorously justified:

- The authors replace the true heuristic distribution $h$ with a tractable proxy distribution $\pi$ within the loss function. This introduces a disconnect between the optimization objective and the actual heuristic behavior.

- When computing the gradient of the loss with respect to scores $s$ in Equation 4, the derivation appears to neglect the fact that the expectation's sampling distribution $h(\hat{y}|\mathcal{G}, s)$ depends on $s$. Standard gradient estimation (like REINFORCE) would require a score function term ($\nabla \log p$), but Equation 5 suggests the derivative is applied only to the inner term. This looks like a mathematical error or an undefined approximation.

2) Limited theoretical analysis: The theoretical results are restricted to existence proofs. Theorem 1 states that for the considered heuristics, there exists a set of parameters $s$ that renders the heuristic optimal. This is a result on expressivity/universality, but the paper lacks statistical learning theory analysis regarding generalization bounds or sample complexity.

3) Fragility of the "probabilistic heuristic" definition: The method relies on sampling distinct solutions $\hat{y}_1, \hat{y}_2$ to create rankings. If the probabilistic heuristic outputs a Dirac distribution (i.e., acts deterministically or the GNN becomes highly confident), the sampled solutions will be identical. In this case, the gradient scaling factor $d(\hat{y}_w, \hat{y}_l)$ becomes zero, potentially causing the gradient to vanish and learning to stall.

4) Arbitrary restriction to linear objectives: The problem statement explicitly restricts the focus to linear objective functions $J_{CO}$. However, the proposed gradient estimator only relies on the scalar values of the objective function to compute the scaling factor, suggesting the framework could theoretically handle any black-box objective.

**Audience:**

Yes

**Audience Explanation:**

Neural combinatorial optimization (NCO) is a significant area of interest within the machine learning community. The paper addresses a key bottleneck in the field: the difficulty of training models without expensive ground-truth optimal solutions. The proposed "hybrid" approach, which integrates GNNs with classical heuristics to ensure feasibility, is a practical direction that practitioners would find relevant. Furthermore, the strong empirical results on the Minimum k-cut problem, where the method improves optimality gaps by an order of magnitude, would be of interest to researchers focusing on graph algorithms and self-supervised learning. The attempt to link preference-based learning (popular in LLMs) to CO is also a novel angle that could inspire future work.

**Broader Impact Concerns:**

I have no concerns on the ethical implications of this work.

**Claims And Evidence:**

No

**Claims Explanation:**

While the empirical evidence demonstrates that the method performs well on the minimum k-cut problem, the mathematical derivation supporting the central contribution (the Preference-Based Gradient Estimation) appears theoretically flawed or significantly under-explained in its current form: see weaknesses above.

**Requested Changes:**

Critical change (required for acceptance):

Clarify the gradient derivation (Equation 4 & 5): The authors must address the mathematical derivation of the gradient. Specifically, they must explain why the dependence of the sampling distribution $h(\hat{y}|\mathcal{G}, s)$ on $s$ is ignored when computing the gradient. Is this a deliberate stop-gradient operation? If so, this must be explicitly stated as a heuristic approximation rather than a derived gradient. If it is not a stop-gradient, the authors must explain the absence of the $\nabla \log p$ term standard in expectations of this form.

Strengthening changes (recommended):

1) Address deterministic collapse: Discuss the behavior of the gradient estimator if the GNN becomes highly confident or the heuristic acts deterministically (outputting a Dirac). For example, by making additional assumptions on the heuristic so that this does not happen (which would contradict the result of theorem 1, however).

2) Expand theoretical analysis: The current Theorem 1 provides an existence proof (universality). It would significantly strengthen the paper to provide insights into generalization bounds or sample complexity to show the method can actually learn these optimal parameters. To do so, studying distributions on graph instances would be interesting.

3) Clarify linearity restriction: The paper restricts $J_{CO}$ to linear objective functions. However, the gradient estimator seems to rely only on scalar evaluations of $J_{CO}$ for ranking. Please clarify if this restriction is technically necessary; if not, broadening the scope to black-box functions would strengthen the contribution.

---

> ### Author Response · Authors · 2025-12-20
> **Rebuttal (Part 1/2)**
>
> We thank reviewer cbsU for their thorough and constructive comments. We believe that their input has allowed us to greatly strengthen the paper, most notably in the theory-focused sections. We now go over each weakness and requested change to address them in detail.
>
> > **Weakness 1.** Unprincipled derivation and approximations: The mathematical derivation of the gradient estimator relies on significant approximations that are not rigorously justified:
> >
> > - The authors replace the true heuristic distribution $h$ with a tractable proxy distribution $\pi$ within the loss function. This introduces a disconnect between the optimization objective and the actual heuristic behavior.
>
> The use of exponential family distributions (such as the one we use for $\pi$) as proxy distributions is a common simplification in the preference learning literature [1, 2]. The particular distribution we chose for $\pi$ is justified here, as it models a distribution proportional to the linear objective function of the CO problem. Of course, this distribution is also useful since it is easy to calculate the gradient of the numerator.
>
> We extended the text after Equation 2, in which $\pi$ is introduced, to include this reasoning behind our choice of proxy distribution. This is in addition to the intuitive explanation already present in the paper.
>
> [1] Rafailov et al., “Direct preference optimization: Your language model is secretly a reward model”. In Advances in Neural Information Processing Systems, 2023.
>
> [2] Meng et al., “SimPO: Simple preference optimization with a reference-free reward”. In Advances in Neural Information Processing Systems, 2024.
>
>
> > **Weakness 1 (continued).**
> >
> >- When computing the gradient of the loss with respect to scores $s$ in Equation 4, the derivation appears to neglect the fact that the expectation's sampling distribution $h( \hat{y} \mid \mathcal{G}, s)$ depends on $s$. Standard gradient estimation (like REINFORCE) would require a score function term ($\nabla \log(p)$), but Equation 5 suggests the derivative is applied only to the inner term. This looks like a mathematical error or an undefined approximation.
>
> > **Requested critical change.** Clarify the gradient derivation (Equation 4 & 5): The authors must address the mathematical derivation of the gradient. Specifically, they must explain why the dependence of the sampling distribution $h(\hat{y} \mid \mathcal{G}, s)$ on $s$ is ignored when computing the gradient. Is this a deliberate stop-gradient operation? If so, this must be explicitly stated as a heuristic approximation rather than a derived gradient. If it is not a stop-gradient, the authors must explain the absence of the $\nabla \log p$ term standard in expectations of this form.
>
> Thank you very much for pointing out this important aspect. In fact, what we missed to mention in the submitted version of the paper, is that we sample the solutions $y_l$ and $y_w$ from the current (old) model, where the parameters $\theta’$ and, therefore, $s’$ are fixed, and use it to train the new model with parameters $\theta$ and $s$. We have now made the corresponding changes in section 5.2. Hence, the $y_w$ and $y_l$ are coming from a different distribution, independent of the parameters $s’$. This is a common approach also used in preference-based learning in the context of LLMs. Thank you again for pointing this out. As previously written in the paper, the derivation was indeed not correct.
>
>
> > **Weakness 2.** Limited theoretical analysis: The theoretical results are restricted to existence proofs. Theorem 1 states that for the considered heuristics, there exists a set of parameters $s$ that renders the heuristic optimal. This is a result on expressivity/universality, but the paper lacks statistical learning theory analysis regarding generalization bounds or sample complexity.
>
> > **Requested strengthening change 2.** Expand theoretical analysis: The current Theorem 1 provides an existence proof (universality). It would significantly strengthen the paper to provide insights into generalization bounds or sample complexity to show the method can actually learn these optimal parameters. To do so, studying distributions on graph instances would be interesting.
>
> Our theorem confirms that there exist GNN outputs for which the optimal solution is found. This shows that even though we use heuristics, we can surpass the heuristic if we combine it with a learned GNN. How close we come to this ideal in practice is shown with our empirical evaluation. We agree that an analysis of generalization bounds or sample complexity would improve the paper, but these proofs are generally extremely hard because it would have to include the analysis of specific optimizers (e.g. SGD, Adam, etc.) and these results are generally extremely rare in the literature of learning theory. We strongly believe this is beyond the scope of our paper.

---

> ### Author Response · Authors · 2025-12-20
> **Rebuttal (Part 2/2)**
>
> > **Weakness 3.** Fragility of the "probabilistic heuristic" definition: The method relies on sampling distinct solutions $\hat{y}_1, \hat{y}_2$ to create rankings. If the probabilistic heuristic outputs a Dirac distribution (i.e., acts deterministically or the GNN becomes highly confident), the sampled solutions will be identical. In this case, the gradient scaling factor $d(\hat{y}_w, \hat{y}_l) becomes zero, potentially causing the gradient to vanish and learning to stall.
>
> > **Requested strengthening change 1.** Address deterministic collapse: Discuss the behavior of the gradient estimator if the GNN becomes highly confident or the heuristic acts deterministically (outputting a Dirac). For example, by making additional assumptions on the heuristic so that this does not happen (which would contradict the result of theorem 1, however).
>
> Towards the end of section 5.2, we describe how we sample multiple solutions and add them to a pool from which the pairs are constructed. This paragraph mentions that we also add solutions obtained from the unmodified heuristic to the pool. This prevents the training from stalling by making it possible for different solutions to occur in the pool. The only case in which training can stall completely is if the unmodified heuristic acts deterministically for all graphs in the training set, which would likely render the heuristic unsuitable regardless.
>
> To clarify this aspect, we added a sentence to the aforementioned paragraph in section 5.2 that highlights how adding solutions from the unmodified heuristic ensures that multiple distinct solutions can always occur in the solution pool, even if the GNN outputs scores that strongly favor one particular solution.
>
>
> > **Weakness 4.** Arbitrary restriction to linear objectives: The problem statement explicitly restricts the focus to linear objective functions $J_{CO}$. However, the proposed gradient estimator only relies on the scalar values of the objective function to compute the scaling factor, suggesting the framework could theoretically handle any black-box objective.
>
> > **Requested strengthening change 3.** Clarify linearity restriction: The paper restricts $J_{CO}$ to linear objective functions. However, the gradient estimator seems to rely only on scalar evaluations of $J_{CO}$ for ranking. Please clarify if this restriction is technically necessary; if not, broadening the scope to black-box functions would strengthen the contribution.
>
> Thank you for bringing this up, we recognize now that this was previously not sufficiently made clear in the paper. The linearity restriction is implicitly used in our choice of the proxy distribution $\pi$, as we explain in our response to weakness 1. To clarify this, we updated the passage in which we introduce $\pi$ to make this connection to the linearity restriction explicit.
>
> The impact of this becomes clear when thinking about the gradients that PBGE outputs; these gradients are only effective for linear objective functions. For non-linear objective functions, the true gradient may look completely different from the estimation that PBGE produces.

---

> > ### Comment · Reviewer_cbsU · 2025-12-21
> > **Answer to rebuttal**
> >
> > I thank the authors for their detailed rebuttal and the revisions made to the manuscript.
> >
> > I have reviewed the updated derivation in Section 5.2. Explicitly denoting the sampling distribution parameters as fixed ($s'$) in Equations 1 and 4 and separating the sampling step in Algorithm 1 successfully addresses the critical mathematical flaw regarding the missing score function term. The derivation is now mathematically consistent with the defined objective. Consequently, I find the responses satisfactory and the changes sufficient to recommend acceptance.
> >
> >
> > However, I would like to offer the following observations regarding the theoretical framing of the method. While these points do not block acceptance, they highlight why the method feels somewhat "unprincipled" despite its empirical success:
> >
> > - You state that you approximate the heuristic distribution $h$ by the proxy $\pi$. It is worth noting that this approximation is applied only within the gradient term (inside the expectation). As per Equation 4, you continue to sample $\hat{y}$ from the original heuristic $h$, not $\pi$. This creates a mismatch: you are optimizing a proxy density $\pi$ using samples drawn from a different distribution $h$.
> >
> > - The derivation relies on several distinct approximations and tricks: freezing the sampling distribution to avoid REINFORCE-like gradients, swapping $h$ for an exponential family proxy $\pi$ to obtain a simple expression (but only in the expectation and not in the sampling). However, the resulting update rule is intuitive: it might have been more transparent to present this simply as a heuristic update rule (a cotangent vector you then backpropagate) rather than framing the method as a "gradient estimation method" of a somewhat obscure loss function derived through multiple layers of approximation.
> >
> > - Furthermore, the paper does not report the evolution of the proposed loss function $\mathcal{L}(\mathcal{D}, s)$ during training. Reporting only downstream metrics (optimality gaps) hides whether the optimization of the surrogate objective is actually stable or convergent, reinforcing the impression that the update rule acts more as an effective heuristic than a principled gradient-based algorithm.
> >
> > - I maintain that the restriction to linear objective functions $J_{CO}$ is not strictly necessary for the framework. Theoretically, one could define $\pi \propto \exp(-J_{CO}(\hat{y}))$ (you then only need the objective function to be differentiable with respect to $s$, with a gradient that can easily be computed). Additionally, while the paper restricts focus to linear objectives, $J_{CO}$ is never rigorously defined in the general problem statement, which adds some ambiguity.
> >
> > - Regarding your response on generalization: "these proofs are generally extremely hard because it would have to include the analysis of specific optimizers (e.g. SGD, Adam)." This is not entirely accurate. One can study the generalization properties of the empirical risk minimizer as the number of data samples grows without analyzing the trajectory of specific optimizers. However, although I think that such questions are the most interesting theoretical questions arising given this framework, I agree that adding such analysis is out of scope for this current work, as the paper focuses on the method and empirical validation.
> >
> >
> > Despite these theoretical reservations, the method is novel, ensures feasibility by design, and the empirical results are compelling. The revisions have resolved the validity of the gradient calculation itself, which was my primary concern.

---

> > > ### Author Response · Authors · 2025-12-22
> > >
> > > Thank you for your reply and the additional feedback. We are glad your critical concern could be resolved, and would like to comment on some of your observations.
> > >
> > > > Furthermore, the paper does not report the evolution of the proposed loss function $\mathcal{L}(\mathcal{D}, s)$ during training. Reporting only downstream metrics (optimality gaps) hides whether the optimization of the surrogate objective is actually stable or convergent, reinforcing the impression that the update rule acts more as an effective heuristic than a principled gradient-based algorithm.
> > >
> > > Thank you for suggesting this, we agree that showing loss curves would be an interesting addition to the paper and will include this in the camera-ready version of the paper.
> > >
> > >
> > > > I maintain that the restriction to linear objective functions $J_\text{CO}$ is not strictly necessary for the framework. Theoretically, one could define $\pi \propto \exp(- J_\text{CO}(\hat{y}))$ (you then only need the objective function to be differentiable with respect to $s$, with a gradient that can easily be computed).
> > >
> > > Thank you for pointing this out; we agree that this can be a useful way to loosen the restriction on the objective function. We view this as an interesting extension to explore in future work.
> > >
> > >
> > > > Additionally, while the paper restricts focus to linear objectives, $J_\text{CO}$ is never rigorously defined in the general problem statement, which adds some ambiguity.
> > >
> > > We updated Section 4 (Problem statement) to include a formal definition of $J_\text{CO}$ and uploaded the revised manuscript.

---

### Review · Reviewer_ht8e · 2025-12-01

**Summary Of Contributions:**

Overall, the paper presents an interesting idea by integrating heuristics into a preference-based training framework, but many key motivations and claimed advantages are not sufficiently validated. The experimental analysis is limited, the baseline coverage is incomplete, and several important concerns about scalability, theory–practice gap, and reliance on heuristics remain unresolved.

**Audience:**

No

**Audience Explanation:**

8. The experiments use small problem sizes such as TSP-100, which is insufficient to demonstrate scalability, especially considering that recent NCO research routinely evaluates significantly larger instances.
9. The evaluation lacks experiments on standard benchmark datasets and relies mainly on synthetic distributions, raising concerns about generalization and practical applicability.
10. The Pareto front in Figure 3 may be incomplete, as it excludes strong solvers like LKH3. In addition, adjusting the sampling budgets of other self-supervised methods could potentially lead them to dominate the proposed method.

**Broader Impact Concerns:**

no suggestion

**Claims And Evidence:**

No

**Claims Explanation:**

1. The paper claims several advantages, but these are not empirically verified; the experiments only compare final solution quality without analyzing whether the proposed benefits actually materialize.
2. The motivation for integrating the heuristic directly into the training loop is not convincingly supported by experiments, and the paper does not clearly demonstrate why this design is preferable to directly using heatmaps with strong decoders such as 2-opt or MCTS.
3. The core contribution, the preference-based gradient estimator, lacks analysis of training stability, hyperparameter sensitivity, ablations, and training cost, making it difficult to assess its robustness or practical usability.
4. The comparison omits many baselines, including preference-based approaches such as BOPO and POCO, heatmap-based self-supervised methods, and other state-of-the-art neural combinatorial optimization (NCO) self-supervised techniques, leading to an incomplete empirical evaluation.
5. From the original POMO results, it appears that POMO still performs better on TSP-100 in both quality and runtime; moreover, POMO is an early and relatively basic self-supervised NCO method, which further highlights the gap in performance. It suggests the proposed method may not yet match strong self-supervised baselines.
6. The method depends heavily on the underlying heuristic, such as random insertion for TSP, which limits solution quality. Since the model only guides the heuristic, the overall performance remains constrained by the heuristic’s inherent strength.
7. The theoretical results are purely existential and do not bridge the gap to practical learnability. It remains unclear whether the model can realistically approximate such heatmaps in real-world scenarios.

**Requested Changes:**

reject

---

> ### Author Response · Authors · 2025-12-20
> **Rebuttal (Part 1/2)**
>
> We thank reviewer ht8e for their review and will now address their points.
>
> > The paper claims several advantages, but these are not empirically verified; the experiments only compare final solution quality without analyzing whether the proposed benefits actually materialize.
>
> We do elaborate on the strengths that are mentioned in the introduction later on in the main body of the paper:
> 1. The introduction claims that our approach improves the heuristic on the instances that occur in practice (i.e. the data distribution). We confirm this by comparing our approach to the unmodified heuristic on both minimum $k$-cut (Tables 1 and 2, as well as Figure 2) and TSP (Table 4, previously called Table 3).
> 2. We claim that incorporating the heuristic into the pipeline ensures that the resulting output is a feasible solution to the CO problem. This is a direct result of the heuristic’s built-in guarantee that its output satisfies the constraints of the CO problem, therefore by using the heuristic’s output as the final output of our pipeline, our approach inherits this guarantee. We added a sentence to section 5.1 that clarifies this point.
> 3. Our introduction claims that once trained, our GNN + heuristic pipeline can act as an improved drop-in replacement for the original heuristic. This is clearly true, as they have the same input and output interface, and we show in our experiments how the performance improves when using the GNN + heuristic pipeline instead of the unmodified heuristic (see e.g. Table 1 and 2, as well as Figure 2)
> 4. We claim that our approach enables fully self-supervised training. Our experiments demonstrate this by training in a self-supervised fashion on both minimum $k$-cut and TSP.
> 5. The introduction states that our approach improves the performance of a commonly used heuristic (Karger–Stein) by an order of magnitude in the case of minimum $k$-cut while only minimally increasing the runtime. This is directly visible from our experimental results in Tables 1 and 2.
>
>
> > The motivation for integrating the heuristic directly into the training loop is not convincingly supported by experiments, and the paper does not clearly demonstrate why this design is preferable to directly using heatmaps with strong decoders such as 2-opt or MCTS.
>
> We do compare against heatmap-based method [1] on TSP, in Table 5 in Appendix F.2 (called Table 4 in the original submission). When comparing between decoders with similar decoding times, our method outperforms [1], even though that approach uses supervised training.
>
> [1] Joshi et al., “An efficient graph convolutional network technique for the travelling salesman problem”, 2019.
>
>
> > The core contribution, the preference-based gradient estimator, lacks analysis of training stability, hyperparameter sensitivity, ablations, and training cost, making it difficult to assess its robustness or practical usability.
>
> We would like to again point out the comprehensive experiments we have conducted to evaluate our approach.
>
> - We show optimality gaps on minimum $k$-cut for $k = 2$, and compare against the unmodified Karger–Stein algorithm as well as multiple supervised and unsupervised baselines in Table 1 (page 7)
> - We also show optimality gaps for larger values of $k$ in Table 2 (page 8)
> - We show how our approach reduces the amount of times that the Karger–Stein algorithm needs to be run in order to find the optimal cut, on both graphs with $n = 100$ and $n = 200$ nodes, in Figure 2 (page 8)
> - In Appendix F.1, we also show this for larger values of $k$ in Figure 4 (page 25)
> - We show the Pareto frontier of learned, self-supervised methods on TSP, based on optimality gap and test runtime in Figure 3 (page 9)
> - We show optimality gaps on TSP for multiple graph sizes, and compare against a wide variety of learned and non-learned baselines, in Table 4 (page 10)
> - In Appendix F.2, we show an extended comparison of optimality gap results on TSP, including additional supervised, self-supervised and non-learned baselines, in Table 5 (page 26)
>
> Page and table numbers refer to the updated manuscript.
>
> To illustrate the training cost, we updated the experiments section with an overview table of the amount of time required for training based on different values of $n$ and $k$ for minimum $k$-cut. These training times are:
> - $n = 100, k = 2$: 4h 8min
> - $n = 100, k = 3$: 7h 39min
> - $n = 100, k = 4$: 6h 13min
> - $n = 200, k = 2$: 15h 46min
>
> As a point of comparison, generating ground truth labels for the supervised baselines on NOIgen+ graphs with $n = 100$ nodes and $k=2$ took 21h for the entire dataset.

---

> ### Author Response · Authors · 2025-12-20
> **Rebuttal (Part 2/2)**
>
> > The comparison omits many baselines, including preference-based approaches such as BOPO and POCO, heatmap-based self-supervised methods, and other state-of-the-art neural combinatorial optimization (NCO) self-supervised techniques, leading to an incomplete empirical evaluation.
>
> We would like to point out that BOPO [2] and POCO [3] were first made public this year, after our paper has been published on arXiv, hence they constitute concurrent work. This is why they do not appear in our evaluation table. We already cite these papers as concurrent work in our related work section.
>
> [2] Liao et al., “BOPO: Neural combinatorial optimization via best-anchored and objective-guided preference optimization”, 2025.
>
> [3] Pan et al., “Preference optimization for combinatorial optimization problems”, 2025.
>
>
> > The method depends heavily on the underlying heuristic, such as random insertion for TSP, which limits solution quality. Since the model only guides the heuristic, the overall performance remains constrained by the heuristic’s inherent strength.
>
> It is true that the quality of the heuristic used during training impacts the quality of the solutions in the solution pool from which the pairs are constructed, and can therefore have an impact on the learning process. However, at test time, the heuristic does not put a hard limit on the final solution quality: It can be seen from our analysis in section 5.3 that the optimal solution can be found as long as the GNN provides good input scores.
>
>
> > The theoretical results are purely existential and do not bridge the gap to practical learnability. It remains unclear whether the model can realistically approximate such heatmaps in real-world scenarios.
>
> The theoretical results provide a sanity check that confirms that there are possible GNN outputs for which the optimal solution is found, without relying on good randomness at test time. How close we come to this ideal in practice is best learned from the empirical evaluation.
>
>
> > The evaluation lacks experiments on standard benchmark datasets and relies mainly on synthetic distributions, raising concerns about generalization and practical applicability.
>
> Using synthetic graphs for evaluation is standard practice in the ML for CO literature. The graph generator we used for TSP is the same one that is used for almost all learned methods for TSP, including all of the baselines we show in the paper. The only non-standard datasets we use are for minimum $k$-cut, as existing datasets for this problem (such as NOIgen [4]) are trivial to solve for GNNs due to a large imbalance in edge weights between edges in the optimal cut and other edges.
>
> [4] Nagamochi et al., “Implementing an efficient minimum capacity cut algorithm”. Mathematical Programming, 1994.
>
>
> > The Pareto front in Figure 3 may be incomplete, as it excludes strong solvers like LKH3. In addition, adjusting the sampling budgets of other self-supervised methods could potentially lead them to dominate the proposed method.
>
> Figure 3 only includes learned, self-supervised methods. Traditional, non-learned approaches for TSP, such as LKH-3, are difficult to beat for learned approaches, as (1) they are specifically tailored towards TSP, whereas many learned approaches, including our method, are applicable to a wide range of CO problems, and (2) they are often written in highly optimized C/C++ code, whereas decoders for learned methods are typically written in Python. For the sake of fairness, we therefore only compare against learned, self-supervised methods in Figure 3.

---

### Review · Reviewer_fBYg · 2025-12-02

**Summary Of Contributions:**

**Summary:**
The paper proposes a way to augment existing heuristics for combinatorial optimization problems with a machine learning component. They specifically target problems that admit a graph-based formulation. Its approach consists of training a graph neural network to predict altered edge weights on the problem's graph representation, over which an existing (fast) heuristic is then executed. By learning to predict the right edge weights, this pipeline can produce higher-quality solutions than the ones obtained by running the heuristic on the original, unmodified graph. One obstacle to this approach is the fact that the heuristics produce discrete solutions. Thus, the gradient of the solution with respect to the predicted edge weights is zero almost everywhere, obstructing gradient-based training. To tackle this issue, the authors propose a novel gradient estimation technique called preference-based gradient estimation (PBGE), which operates by comparing the quality of multiple solutions sampled using the heuristic, and which can be used in a self-supervised learning setup. The authors evaluate the performance of their technique on two CO problems: the travelling salesman problem (TSP) and the minimum k-cut problem.


**Strengths:**

* The paper introduces an interesting method for parameterizing heuristics in a generic, heuristic-agnostic manner by modifying the input graph on which the heuristic operates.

* To train the model that predicts the input modifications, the paper introduces a novel gradient estimation scheme.

* The paper is well written, clearly structured and pleasant to read.

* The experimental results are quite convincing (see below).

**Weaknesses:**

* The approach is only applicable to graph-based problems where the decision variables represent edge selections.

* The approach is only applicable when using probabilistic heuristics.

I also have some questions that I would like the authors to address, as well as several requests for changes to the paper. Please see the sections below.

**Additional Comments:**

* **Q8**: Why is the $-1$ term included in the scaling factor? Is this important?

* **Q9**: I see a clear connection between PBGE and the pairwise learning-to-rank approach (e.g., used in [3] in a decision-focused learning context), and particularly its best-vs-rest variant. This makes me wonder: could the use of other ranking approaches also work in the setting of this paper? For example, could a listwise approach be used? I expect that this would come at no additional cost, since it could be applied over the same solution pools currently used for PBGE. Is this something you have experimented with? What are your thoughts?

* **Q10**: I find it surprising that, by design, the difference in solutions $\hat{y}_1$ and $\hat{y}_2$ may only arise from randomness inherent to the heuristic used. If my understanding is correct, $\hat{y}_1$ and $\hat{y}_2$ differ only when two runs of the heuristic on the same scores produce different solutions. I am curious why this design is chosen. Does this not make the approach reliant the use of a sufficiently stochastic heuristic? An alternative could be to have the GNN predict a Gaussian distribution over outputs $s$ (using the reparameterization trick), and to run the heuristic on these perturbed scores. In addition to removing the reliance on stochastic heuristics, this could also provide more control over the difference between $\hat{y}_1$ and $\hat{y}_2$ (through the size of the Gaussian’s variance). Would this also be a valid approach? What are your thoughts?

[3] Mandi, J., Bucarey, V., Tchomba, M. M. K., & Guns, T. (2022). Decision-focused learning: Through the lens of learning to rank. In *International conference on machine learning* (pp. 14935-14947). PMLR.

**Audience:**

Yes

**Audience Explanation:**

This paper will certainly be of interest to members of TMLR’s audience, particularly readers interested in intersections between combinatorial optimization and machine learning. I myself found the paper very interesting, and especially its approach of indirectly parameterizing heuristics by having a machine learning component alter the input graph.

**Broader Impact Concerns:**

I have no broader impact concerns.

**Claims And Evidence:**

Yes

**Claims Explanation:**

The results are quite convincing. The authors compared their approach against standard heuristics, as well as heuristics parameterized according to their proposed scheme but trained using methods other than PBGE (e.g., I-MLE, BCE, REINFORCE). They experiment on two problems: the minimum k-cut problem and the TSP. On top of the base tables of results (Table 1 and Table 3), some very welcome additional insights are provided. On the minimum k-cut problem, we are given insight into how the proportion of graphs for which the optimal cut is found changes when the number of runs of the heuristic increases, for different training methods. We are also given results for varying values of *k.* For the TSP, an extensive list of baselines is used for comparison, and the results are nicely visualized along with a Pareto frontier in Figure 3.

However, while I generally like the exposition of the results, I also have some questions:

* **Q1:** What is the size of the test sets? Very little information is given in general about the dataset size and the training-validation-test splits used. All I could find on this matter is given in appendix G, but here too, test set sizes are not reported.

* **Q2:** Why is the REINFORCE baseline only used on the TSP, and not on the minimum k-cut problem? Presently, no motivation is given for this decision.

* **Q3:** Similarly, the supervised baselines are only used on the minimum k-cut problem, and not on the TSP, without a clear motivation given. What is the reasoning behind this decision?

* **Q4:** It is unclear to me what the “best out of 3 runs” columns in Table 1 contribute to the insight we are supposed to draw from the table. This is not explicitly touched upon in the accompanying text either.

* **Q5:** For the supervised baselines in Table 1, is the time required to compute the optimal solution (for supervision) included in the reported computation times?

* **Q6:** What do the runs in Table 3 (e.g., “I-MLE (random ins., 20 runs)”) refer to? Are these different runs, of which the best one is selected and reported?

* **Q7:** I feel like the claim “I-MLE acts as a representative of the decision-focused learning paradigm” needs some backing. Why is I-MLE chosen? Would other approaches, like the perturbed optimizers from [1] or the negative identity with projection from [2], also work?

[1] Berthet, Q., Blondel, M., Teboul, O., Cuturi, M., Vert, J. P., & Bach, F. (2020). Learning with differentiable pertubed optimizers. *Advances in neural information processing systems*, *33*, 9508-9519.

[2] Sahoo, S. S., Paulus, A., Vlastelica, M., Musil, V., Kuleshov, V., & Martius, G. Backpropagation through Combinatorial Algorithms: Identity with Projection Works. In *The Eleventh International Conference on Learning Representations*.

**Requested Changes:**

* If there are no clear motivations behind the decisions that I refer to in questions **Q2**, **Q3**, and **Q7**, I feel like these matters should be addressed by adding the appropriate experiments.

* More information on the size of the datasets and the train-validation-test splits must be given in the main text.

* I would like to see a more refined discussion in the introduction of which class of problems this approach can be applied to exactly.  Only when encountering Equation 2, in which the inner product of decision variables $\hat{y}$ and GNN scores $s$ is taken, I realized that the proposed approach is not applicable to all graph problems, but only to graph problems where the decision variables represent edge selections. So, for example, the maximum weight independent set problem, cannot be tackled, since its decision variables represent node selections, rather than edge selections. This understanding was only explicitly confirmed in the future work section, at the very end of the paper. This feels somewhat misleading. I feel that this should be cleared up much earlier on in the paper.

* In Section 5, the discussion of the solution pools and what happens at test time now fall under the “Gradient scaling.” paragraph. I suggest to separate them, by giving them their own named paragraphs, e.g., “Solutions pools.” for the former, and “Test time.” for the latter.

* I really like the intuitive explanation of the gradient, given right under Equation 5. I would like to similarly see an intuitive explanation of loss as well, under Equation 1. The way I understand it, the loss is large when the probability of the ‘loser’ (i.e., $\hat{y}_l$) in the numerator is large compared to the probability of the ‘winner’ (i.e., $\hat{y}_w$) in the denominator. Some clarification of this in the text would be appreciated.

* At the end of the introduction, the authors write: “Our approach improves the performance of a commonly used heuristic by an order of magnitude in the case of minimum k-cut while only minimally increasing the runtime.” Strangely, a similar sentence summarizing the results on the TSP is not given. I suggest to add such a sentence, perhaps referring to the fact that PBGE achieves Pareto-optimality on the TSP.

* There is currently a typo in the theorem (“let and let”).

* Currently, “combinatorial optimization” is still used in full after the introduction of the acronym “CO”. Similarly, both forms “i.e., …” and “i.e. …” are used. I would appreciate another pass over the paper to resolve these minor inconsistencies.

---

> ### Author Response · Authors · 2025-12-20
> **Rebuttal (Part 1/4)**
>
> We would like to thank reviewer fBYg for their detailed and insightful review, which has allowed us to make many improvements throughout the paper. In the following we go over each question and requested change and respond to them in detail.
>
> > **Question 1.** What is the size of the test sets? Very little information is given in general about the dataset size and the training-validation-test splits used. All I could find on this matter is given in appendix G, but here too, test set sizes are not reported.
>
> Our datasets contain 10,000 training graphs, 1,000 validation graphs and 1,000 test graphs. We agree that this information should be in the main paper, so we added a corresponding sentence to section 6.1.
>
>
> > **Question 2.** Why is the REINFORCE baseline only used on the TSP, and not on the minimum k-cut problem? Presently, no motivation is given for this decision.
>
> We ran self-supervised experiments with REINFORCE for minimum $k$-cut and obtained the following results:
>
> On unweighted graphs:
>
> | Method | Single run | Best out of 3 |
> |:--------------------------|:------------------:|:-----------------:|
> | **Self-supervised** |                       |                       |
> | REINFORCE          | 2.18% ± 0.15 | 0.41% ± 0.11 |
>
> On NOIgen+:
>
> | Method | Single run | Best out of 3 |
> |:--------------------------|:------------------:|:-----------------:|
> | **Self-supervised** |                       |                       |
> | REINFORCE          | 3.15% ± 0.36 | 0.18% ± 0.03 |
>
> A corresponding row was added to Table 1.
>
> REINFORCE performs better than I-MLE, but worse than PBGE across both datasets. This is consistent with the results we found on TSP.
>
>
> > **Question 3.** Similarly, the supervised baselines are only used on the minimum k-cut problem, and not on the TSP, without a clear motivation given. What is the reasoning behind this decision?
>
> Table 5 in Appendix F (called Table 4 in the original submission) contains extended results for TSP, including the supervised BCE baseline (though the baselines listed in section 6.1 only include the ones for which results are shown in the main paper). We additionally ran the supervised I-MLE baseline for TSP and added the results to Table 5. The results were as follows. The table also includes the supervised BCE results for your convenience, copied from our appendix.
>
> | Method (decoder in parentheses) |      n = 20      |       n = 50        |       n = 100       |
> |:---------------------------------------------|:-----------------:|:-------------------:|:--------------------:|
> | **Supervised**                              |                       |                         |                          |
> | BCE (random ins., 20 runs)          | 0.15% ± 0.01 |   0.95% ± 0.03 |   2.86% ± 0.04 |
> | BCE (random ins., 100 runs)        | 0.04% ± 0.00 |   0.59% ± 0.02 |   1.75% ± 0.03 |
> | I-MLE (random ins., 20 runs)        | 2.67% ± 0.06 | 14.71% ± 0.18 | 26.57% ± 0.17 |
> | I-MLE (random ins., 100 runs)      | 1.17% ± 0.04 |   9.66% ± 0.10 | 18.91% ± 0.11 |
>
> Supervised I-MLE performs much worse than a simple BCE loss, which is expected, since the additional indirection of running the GNN outputs through the heuristic makes training more difficult, and I-MLE generally does not provide good gradients in the CO problems we experimented on. It also performs worse than self-supervised PBGE, but better than self-supervised I-MLE. All of these findings are consistent with what we have seen on minimum $k$-cut.
>
> We are open to also including both baselines in Table 4 in the main document if this improves clarity.
>
> The introduction of the baselines in section 6.1 has been adjusted to reflect the additional experiments.
>
>
> > **Question 4.** It is unclear to me what the “best out of 3 runs” columns in Table 1 contribute to the insight we are supposed to draw from the table. This is not explicitly touched upon in the accompanying text either.
>
> Thank you for pointing out this potential source of confusion. For minimum $k$-cut, it is common to run the Karger–Stein algorithm repeatedly and use the best result. It is therefore interesting to see how the result improves for multiple Karger–Stein runs. This is somewhat hinted at in the final sentence of section 5.2, but should have been made much clearer. We therefore added an explaining sentence in the paragraph that discusses Table 1.

---

> ### Author Response · Authors · 2025-12-20
> **Rebuttal (Part 2/4)**
>
> > **Question 5.** For the supervised baselines in Table 1, is the time required to compute the optimal solution (for supervision) included in the reported computation times?
>
> No; the timings only measure the calculations made at test time. This includes GNN inference and running the Karger–Stein algorithm to decode the solution. The timings are similar within each column since they are dominated by the Karger–Stein algorithm.
>
> As for the calculation of the ground truth solutions: For the graphs with edge weigths (NOIgen+ graphs), calculating the ground truth solution took 35 seconds per graph on average, using the same hardware that was used for the other timings in the paper. Using parallelization, computing ground truth solutions for the entire dataset took around 21h. The graphs without edge weights were constructed in such a way that the optimal cut is known from construction with very high likelihood (see Appendix E.1 for details). In fact, we haven’t encountered a single graph in the dataset where a better cut was found. Each graph still contains many cuts that are only slightly larger than the optimal cut, keeping the problem challenging.
>
>
> > **Question 6.** What do the runs in Table 3 (e.g., “I-MLE (random ins., 20 runs)”) refer to? Are these different runs, of which the best one is selected and reported?
>
> “20 runs” means that the decoder was run 20 times using the same GNN outputs as input parameters, and the best of these 20 tours is used as the final result. To be clear: this does not refer to 20 training runs or 20 GNN evaluations. This procedure is briefly described at the end of section 5.2, but we recognize now that this warrants further explanation. We therefore added a clarifying sentence to the paragraph that discusses Table 4 (called Table 3 in the original submission).
>
>
> > **Question 7.** I feel like the claim “I-MLE acts as a representative of the decision-focused learning paradigm” needs some backing. Why is I-MLE chosen? Would other approaches, like the perturbed optimizers from [1] or the negative identity with projection from [2], also work?
> >
> > [1] Berthet, Q., Blondel, M., Teboul, O., Cuturi, M., Vert, J. P., & Bach, F. (2020). Learning with differentiable pertubed optimizers. Advances in neural information processing systems, 33, 9508-9519.
> >
> > [2] Sahoo, S. S., Paulus, A., Vlastelica, M., Musil, V., Kuleshov, V., & Martius, G. Backpropagation through Combinatorial Algorithms: Identity with Projection Works. In The Eleventh International Conference on Learning Representations.
>
> I-MLE is chosen because it generalizes [A] (a precursor to [2]) and [1]. It performs a general form of perturbed optimization, where the perturbation is done with arbitrary noise distributions and by the gradient of an upstream loss. Please note that we do not intend to say that I-MLE is the best method among all alternatives, but one of several representative methods. Hence, [1, 2] can be applied here and we updated the manuscript to mention this.
>
> [A] Marin Vlastelica, Anselm Paulus, Vít Musil, Georg Martius, and Michal Rolínek. Differentiation of blackbox combinatorial solvers. In International Conference on Learning Representations, May 2020.
>
>
> We also ran additional experiments that replace the PBGE gradient estimation with [2], and obtained the following results.
>
> On minimum $k$-cut, unweighted graphs:
>
> | Method | Single run | Best out of 3 |
> |:------------------------------|:------------------:|:------------------:|
> | **Self-supervised**      |                       |                        |
> | Identity with projection | 3.62% ± 0.15 | 0.61% ± 0.05 |
>
> On minimum $k$-cut, NOIgen+:
>
> | Method | Single run | Best out of 3 |
> |:------------------------------|:------------------:|:------------------:|
> | **Self-supervised**      |                       |                        |
> | Identity with projection | 7.46% ± 0.40 | 0.42% ± 0.07 |
>
> On TSP:
>
> | Method (decoder in parentheses)                       |    n = 20    |    n = 50    |    n = 100    |
> |:---------------------------------------------------------------|:--------------:|:-------------:|:---------------:|
> | **Self-supervised**                                             |                   |                  |                    |
> | Identity with projection (random ins., 20 runs)   | 8.92% ± 0.13 | 31.87% ± 0.32 | 56.55% ± 0.33 |
> | Identity with projection (random ins., 100 runs) | 5.16% ± 0.09 | 17.85% ± 0.09 | 39.69% ± 0.28 |
>
> We added these numbers to Tables 1 and 4, respectively, and added the baseline to our list of baselines in section 6.1.
>
> These results are very similar to those of the I-MLE baseline, with slightly worse optimality gaps on minimum $k$-cut and slightly better optimality gaps on TSP. PBGE outperforms both of these baselines on both problems.

---

> ### Author Response · Authors · 2025-12-20
> **Rebuttal (Part 3/4)**
>
> > **Requested Change 1.** If there are no clear motivations behind the decisions that I refer to in questions **Q2**, **Q3**, and **Q7**, I feel like these matters should be addressed by adding the appropriate experiments.
>
> We ran additional experiments in order to address these points. Please see our comments on Q2, Q3, and Q7 for the corresponding results.
>
>
> > **Requested Change 2.** More information on the size of the datasets and the train-validation-test splits must be given in the main text.
>
> We added the following sentence to section 6.1: “For each dataset, we generate 10,000 training graphs, 1,000 validation graphs and 1,000 test graphs.”
>
>
> > **Requested Change 3.** I would like to see a more refined discussion in the introduction of which class of problems this approach can be applied to exactly. Only when encountering Equation 2, in which the inner product of decision variables $\hat{y}$ and GNN scores $s$ is taken, I realized that the proposed approach is not applicable to all graph problems, but only to graph problems where the decision variables represent edge selections. So, for example, the maximum weight independent set problem, cannot be tackled, since its decision variables represent node selections, rather than edge selections. This understanding was only explicitly confirmed in the future work section, at the very end of the paper. This feels somewhat misleading. I feel that this should be cleared up much earlier on in the paper.
>
> Thank you for pointing this out, we agree that this needs to be stated early on in the paper. We updated the introduction to clarify that our work focuses on edge selection problems.
>
>
> > **Requested Change 4.** In Section 5, the discussion of the solution pools and what happens at test time now fall under the “Gradient scaling.” paragraph. I suggest to separate them, by giving them their own named paragraphs, e.g., “Solutions pools.” for the former, and “Test time.” for the latter.
>
> Thank you for this suggestion. We think this change improves the structure of this section and we separated the two short passages into their own titled paragraphs as requested.
>
>
> > **Requested Change 5.** I really like the intuitive explanation of the gradient, given right under Equation 5. I would like to similarly see an intuitive explanation of loss as well, under Equation 1. The way I understand it, the loss is large when the probability of the ‘loser’ (i.e., $\hat{y}_l$) in the numerator is large compared to the probability of the ‘winner’ (i.e., $\hat{y}_w$) in the denominator. Some clarification of this in the text would be appreciated.
>
> Yes, your intuition is correct. We agree that conveying an intuitive understanding of the loss would make this section easier to read. We therefore updated the text below Equation 1 to read as follows.
>
> “Intuitively, this rewards assigning a high probability to the better solution $\hat{\boldsymbol{y}}_w$ and a low probability to the worse solution $\hat{\boldsymbol{y}}_l$. $d(\hat{\boldsymbol{y}}_w, \hat{\boldsymbol{y}}_l)$ is a scaling factor; as we will see later, its purpose is to scale the gradients based on the distance between the objective values of $\hat{\boldsymbol{y}}_w$ and $\hat{\boldsymbol{y}}_l$.”
>
> The changes from the previous version lie mostly in the first sentence. We also updated the text before Equation 1 to indicate that $\hat{\boldsymbol{y}}_w$ and $\hat{\boldsymbol{y}}_l$ stand for “winner” and “loser”, respectively, for the benefit of readers who are not familiar with this convention from the preference learning literature and who might otherwise be confused why these specific subscripts were chosen.
>
>
> > **Requested Change 6.** At the end of the introduction, the authors write: “Our approach improves the performance of a commonly used heuristic by an order of magnitude in the case of minimum k-cut while only minimally increasing the runtime.” Strangely, a similar sentence summarizing the results on the TSP is not given. I suggest to add such a sentence, perhaps referring to the fact that PBGE achieves Pareto-optimality on the TSP.
>
> Thank you for suggesting this; we added the following sentence to the introduction. “On TSP, our approach is competitive with existing approaches and achieves Pareto-optimality.”
>
>
> > **Requested Change 7.** There is currently a typo in the theorem (“let and let”).
>
> Thank you for pointing this out, this has been fixed in the updated manuscript.

---

> ### Author Response · Authors · 2025-12-20
> **Rebuttal (Part 4/4)**
>
> > **Requested Change 8.** Currently, “combinatorial optimization” is still used in full after the introduction of the acronym “CO”. Similarly, both forms “i.e., …” and “i.e. …” are used. I would appreciate another pass over the paper to resolve these minor inconsistencies.
>
> We went through the document again and fixed these smaller issues. We decided to keep the long version of “combinatorial optimization (CO)” in its definition in the background section, to remind the reader of what is being defined. Similarly, we also kept the long version of TSP in its description.
>
>
> > **Question 8.** Why is the $-1$ term included in the scaling factor? Is this important?
>
> Yes, the $-1$ term is important. During training, the heuristic might find two distinct solutions of equal quality. In this case, there would be some non-zero gradient at the edges where the two solutions differ, but the direction of the gradient would be meaningless. To avoid this, we would like to scale the gradient to zero. Since in this case, the ratio between the solution qualities is 1, subtracting 1 sets the gradient to 0. Similarly, if the difference in quality between the two solutions is almost zero, we would like the gradient to be almost zero etc.
>
> The paragraph in question already contained a sentence that hints at this: “In particular, if the two solutions are of the same quality, the gradient is set to zero, so we do not move the GNN towards either solution.” We now adjusted this sentence to highlight the connection to the $-1$ term, as this did not become clear previously.
>
>
> > **Question 9.** I see a clear connection between PBGE and the pairwise learning-to-rank approach (e.g., used in [3] in a decision-focused learning context), and particularly its best-vs-rest variant. This makes me wonder: could the use of other ranking approaches also work in the setting of this paper? For example, could a listwise approach be used? I expect that this would come at no additional cost, since it could be applied over the same solution pools currently used for PBGE. Is this something you have experimented with? What are your thoughts?
> >
> > [3] Mandi, J., Bucarey, V., Tchomba, M. M. K., & Guns, T. (2022). Decision-focused learning: Through the lens of learning to rank. In International conference on machine learning (pp. 14935-14947). PMLR.
>
> Thank you for pointing out this interesting direction! Yes, a listwise approach should be applicable to our method, though we have not experimented with such an approach. We think that this constitutes an interesting avenue for future research.
>
>
> > **Question 10.** I find it surprising that, by design, the difference in solutions and may only arise from randomness inherent to the heuristic used. If my understanding is correct, and differ only when two runs of the heuristic on the same scores produce different solutions. I am curious why this design is chosen. Does this not make the approach reliant the use of a sufficiently stochastic heuristic? An alternative could be to have the GNN predict a Gaussian distribution over outputs (using the reparameterization trick), and to run the heuristic on these perturbed scores. In addition to removing the reliance on stochastic heuristics, this could also provide more control over the difference between and (through the size of the Gaussian’s variance). Would this also be a valid approach? What are your thoughts?
>
> The approach you propose is theoretically valid. We have experimented with a similar approach that makes deterministic heuristics probabilistic by adding noise to their input (i.e. to the GNN output scores). However, we have since abandoned this direction as it didn’t achieve good results.

---

### Comment · Action_Editor_pci4 · 2025-12-16

Dear authors,
Please remember to answer reviewer comments as early as you can, ideally with a revised version of the paper. The discussion period has been extended at your request, and will last for one more week.
Best,
Your Action Editor

---

### Author Response · Authors · 2025-12-20
**Summary of Changes**

We sincerely thank each of the reviewers for their thoughtful feedback and questions. In addition to our direct responses to the reviews, we updated the manuscript based on the points raised by the reviewers. The updated PDF highlights all changes in red for easy reference.

In summary, these are the changes to the manuscript, and the reviewer comments that they relate to:

- **Section 1 (Introduction)**: Clarified that our approach is only applicable to edge-selection problems (reviewer fBYg, requested change 3)
- **Section 1 (Introduction)**: Briefly described the results on TSP (reviewer fBYg, requested change 6)
- **Section 5.1 (Parameterizing heuristics)**: Added a sentence to explain that the GNN + heuristic pipeline inherits the guarantee from the heuristic that the final output meets the CO problem’s constraints (reviewer ht8e, comment 1)
- **Section 5.2 (Preference-based gradient estimation)**: Corrected the gradient derivation by stating that $y_w, y_l$ are sampled from the old model with fixed parameters $\theta’$ (reviewer cbsU, weakness 1 and requested critical change)
- **Section 5.2 (Preference-based gradient estimation)**: Added an intuitive explanation of the loss. Also clarified what the subscripts in $y_w$ and $y_l$ stand for (reviewer fBYg, requested change 5)
- **Section 5.2 (Preference-based gradient estimation)**: Added justification of the proxy distribution $\pi$, while making its connection to the restriction to linear CO objective functions explicit (reviewer cbsU, weakness 1, weakness 4 and requested strengthening change 3)
- **Section 5.2 (Preference-based gradient estimation)**: In the explanation of the scaling factor, made the purpose of the $-1$ term more explicit (reviewer fBYg, question 8)
- **Section 5.2 (Preference-based gradient estimation)**: Improved the structure of the “gradient scaling” paragraph by splitting it into multiple named paragraphs (reviewer fBYg, requested change 4)
- **Section 5.2 (Preference-based gradient estimation)**: Explain how the way we build the solution pool prevents the training from stalling (reviewer cbsU, weakness 3 and requested change 1)
- **Section 5.3 (Theoretical analysis)**: Fixed a typo in Theorem 1 (reviewer fBYg, requested change 7)
- **Section 6.1 (Problem instance generation and baselines)**: Added information on dataset sizes (Reviewer fBYg, question 1 and requested change 2)
- **Section 6.1 (Problem instance generation and baselines)**: Updated the list of baselines to reflect the new experiments. Noted that other decision-focused-learning approaches are also applicable (Reviewer fBYg, questions 2, 3, and 7, as well as requested change 1)
- **Section 6.2 (Experimental results), Table 1**: Added self-supervised experiments using the REINFORCE baseline and the identity with projection baseline (reviewer fBYg, question 2, question 7 and requested change 1)
- **Section 6.2 (Experimental results)**: Explained why we include evaluations where Karger–Stein is run 3 times (reviewer fBYg, question 4)
- **Section 6.2 (Experimental results)**: Added an overview table (Table 3) of the amount of time required to train on minimum $k$-cut using PBGE, for different graph sizes and values of $k$ (reviewer ht8e, comment 3)
- **Section 6.2 (Experimental results), Table 4 (called Table 3 in the original submission)**: Added self-supervised experiments using the identity with projection baseline (reviewer fBYg, question 7 and requested change 1)
- **Section 6.2 (Experimental results)**: Better explained the decoder “Random ins. 20/100 runs” (reviewer fBYg, question 6)
- **Appendix F.2 (Extended comparison for TSP), Table 5 (called Table 4 in the original submission)**: Added supervised experiments using the I-MLE baseline (reviewer fBYg, question 3 and requested change 1)
- **Throughout the paper**: More consistent usage of the short forms of acronyms after introducing the long form. Also, corrected minor punctuation errors (reviewer fBYg, requested change 8)

We apologize for taking so long to reply to the reviews. The rebuttal process started on the same day as NeurIPS, which I attended to present this paper at a workshop. Unfortunately I became ill after returning from NeurIPS, which has cost additional time (though I am on my way to recovery now). We would like to thank the action editor for graciously granting an extension to the discussion period to compensate for the overlap with NeurIPS.

---

### Comment · Action_Editor_pci4 · 2026-04-01
**Camera-ready: final edit**

Thank you for preparing the camera-ready version of the paper. All my remarks have been taken into account, with the exception of the first one on the scope of the contributions. To clarify, are the authors claiming that plugging predicted edge weights into a probabilistic heuristic is a novel method that they introduce? If I remember correctly, this idea goes back as far as https://arxiv.org/abs/2006.10643 and probably even further.

---

> ### Author Response · Authors · 2026-04-02
>
> Dear Guillaume, thank you for the comment. We updated the list of contributions and submitted a new version.

---

> > ### Comment · Action_Editor_pci4 · 2026-04-03
> >
> > Thank you, that is better. Please fix the typo added in the second contribution and I'll validate the camera-ready:
> >
> > > 2. An extensive experimental evaluation **of thereof** on common CO problems.

---

> > > ### Author Response · Authors · 2026-04-05
> > >
> > > Sorry for that small oversight, should be fixed now.

---

### Decision · Action_Editor_pci4 · 2026-02-04

**Recommendation:** Accept with minor revision

**Additional Comments:**

Congratulations on paper acceptance!
Here are the changes required for the camera-ready version, along with the reviewer who suggested them (when appropriate):

1. Rephrase the contributions in section 1. As far as I can tell, contributions 1 and 2 are already known in the field (defining new edge weights --or probabilities-- with a learnable GNN has been common for some time now), while contributions 3 and 4 are the true heart of the paper.
2. Extend the limitations section. The review and rebuttal phase surfaced many interesting questions, which deserve to be mentioned:
    - possible generalization to nonlinear objectives (cbsU)
    - role of the successive mathematical approximations used to obtain the gradient (cbsU)
    - strong impact of heuristic choice (ht8e) and limitation to probabilistic heuristics (fBYg)
    - theoretical perspectives on learnability, as opposed to the given theorems which only tackle expressivity (cbsU, ht8e)
    - potential use of other ranking approaches beyond pairwise comparison (fBYg)
    - missing evaluation on larger instances (ht8e)
3. Add loss curves to the experiments section (cbsU)
4. Perform a last typo check:
    - page 6: "a scaling factor is used to weigh [not weight] important gradients"
    - page 16: "Since a Hamiltonian cycle [not cycles] to a given graph"
    - etc.

**Audience:**

Yes

**Audience Explanation:**

Learning-enhanced combinatorial optimization is a thriving area of research, and self-supervised methods are the most promising ones since they do not require expensive ground truth solutions to NP-hard problems. The proposed PBGE algorithm is intuitive and straightforward to implement, which makes it a solid baseline for future research.

**Claims And Evidence:**

Yes

**Claims Explanation:**

This paper introduces a new preference-based gradient estimator for discrete optimization solvers, called PBGE. It exhibits strong empirical performance on two problems of interest. Experimental evaluation is thorough, albeit limited to small problems, and some theoretical insights (related to expressivity) are also given.